# Multifunctional and biodegradable self-propelled protein motors

Abdon Pena-Francesch [1], Joshua Giltinan[1] & Metin Sitti [1]

A diversity of self-propelled chemical motors, based on Marangoni propulsive forces, has been developed in recent years. However, most motors are non-functional due to poor performance, a lack of control, and the use of toxic materials. To overcome these limitations, we have developed multifunctional and biodegradable self-propelled motors from squid-derived proteins and an anesthetic metabolite. The protein motors surpass previous reports in performance output and efficiency by several orders of magnitude, and they offer control of their propulsion modes, speed, mobility lifetime, and directionality by regulating the protein nanostructure via local and external stimuli, resulting in programmable and complex loco-motion. We demonstrate diverse functionalities of these motors in environmental remediation, microrobot powering, and cargo delivery applications. These versatile and degradable protein motors enable design, control, and actuation strategies in microrobotics as modular propulsion sources for autonomous minimally invasive medical operations in biological environments with air-liquid interfaces.

---

[1] Physical Intelligence Department, Max Planck Institute for Intelligent Systems, 70569 Stuttgart, Germany. Correspondence and requests for materials should be addressed to M.S. (email: sitti@is.mpg.de)

Chemical motors are ubiquitous in nature, ranging from molecular scale (e.g., myosin or kinesin)[1] to micro/milli-meter scale (e.g., locomotion of arthropods)[2]. Some semi-aquatic insects (*Microvelia* and *Velia*) can rapidly move on the water surface, i.e., at the air–water interface, by secreting surfactants that reduce the local surface tension behind them (achieving speeds twice their peak walking speed)[2,3]. This anisotropic surface tension gradient generates a propulsive force towards high surface tension, known as Marangoni propulsion[4]. This propulsion mechanism has inspired the development of materials and artificial motors for self-propelled systems, from active microdroplet swimmers to milli/centimeter-scale robotic platforms[5–10]. Various strategies have been explored for the design of Marangoni motors, where chemical fuel was directly added on the liquid interface or encapsulated inside the motor body[7,8,11,12]. However, self-propelled motor systems have significant limitations that constrain their overall performance: low efficiency (i.e., high volume of fuel is required for sustained locomotion), short mobility lifetime (due to a finite amount of fuel), lack of locomotion direction control (i.e., random non-directional locomotion), and difficult miniaturization (size is limited by fuel storage capacity and fuel-friendly fabrication processes)[13]. Despite recent research on highly absorbent materials for improved fuel storage, such as polymer hydrogels[12,14,15] and metal-organic frameworks[16–18], most self-propelled surface motors in the literature are non-functional, with low efficiency, low to moderate mobility lifetime and speeds, and uncontrolled random locomotion[7,13]. Furthermore, the materials of the motor, the fuel, and/or the swimming media might be toxic to physical and biological environments, posing biocompatibility, biodegradability, and sustainability challenges that need to be addressed to expand the applications of such motor systems to real world scenarios.

Proteins have multiple advantages over synthetic materials: they self-fold into complex hierarchical nanostructures, have programmable physical and chemical properties, and have improved biocompatibility and biodegradability[19]. In this work, we developed chemical motors from squid ring teeth (SRT) proteins (or suckerins), which are structural proteins from predatory appendages in the suction cups of squids[20]. SRT proteins have a repetitive segmented amino acid sequence with alternating amorphous and crystal-forming domains that self-assemble into β-sheet-stabilized protein networks[21]. These β-sheet nanocrystals act as physical and reversible cross-links via hydrogen bonding, bestowing programmable and dynamic mechanical, thermal, optical, and conducting properties on the material[22]. Due to the reversible nature of β-sheet nanocrystals, SRT proteins can be processed by a wide range of nanofabrication and microfabrication methods[23], enabling the fabrication of protein-based materials (in this case, self-propelled motors) across length scales. Complementing the dynamic properties of SRT proteins, we use hexafluoroisopropanol (HFIP), a highly versatile solvent with very low surface tension, as fuel to generate propulsive Marangoni forces[24]. HFIP is a metabolite of sevoflurane, a FDA-approved and widely used inhalation general anesthetic[25]. Therefore, the pharmacokinetics of HFIP are well-documented: ~5% of a sevoflurane dose is metabolized into HFIP, which is then rapidly conjugated with glucuronic acid and excreted in urine[25]. Moreover, intravenous administration of HFIP has been shown to attenuate inflammation and improve survival in murine septic peritonitis in several clinical studies[26,27].

Here, we present multifunctional and biodegradable self-propelled motors from squid-derived proteins (SRT) and an anesthetic metabolite (HFIP) that outperform previous Marangoni motor systems and overcome previous material and performance limitations. Due to the tunable structure of SRT proteins and their unique interaction with HFIP fuel, our protein-based motors overcome major limitations in the self-propelled robotics field: orders of magnitude higher performance output and efficiency than the previously reported studies, high mobility lifetime, miniaturizable fabrication, controlled, and programmable locomotion (including active direction control by magnetic steering and locomotion after the chemical fuel is exhausted), on-demand termination and degradation, and biocompatibility/biodegradability. Additionally, we demonstrate different functionalities of the protein motors, such as modular powering of microrobots at the air–water interface, removal of water contaminants for environmental remediation, and targeted cargo delivery via motor self-destruction.

## Results

**Motor fabrication, design, and propulsion mechanism.** In order to leverage the tunable physical and chemical properties of SRT, we developed a facile microfabrication process to manufacture protein-based motors capable of complex locomotion (Fig. 1a). First, SRT protein is obtained from either natural (waste product of fisheries) or biosynthetic sources (protein expression in genetically modified organisms)[23]. In this work, native SRT proteins were extracted from the suction cups of *Loligo vulgaris* squid, purified, and dissolved in HFIP to form a motor solution (to which dyes were added for enhanced visualization). The motor solution was cast on soft non-adhesive substrates and the HFIP solvent was partially evaporated (protein chains self-assemble into a solid film, Supplementary Fig. 1). The resulting protein films (20 μm in thickness) were cut by laser micro-machining to specified motor designs (Supplementary Fig. 2), and the motors were then transferred for experiments or stored for later use. We adopted the low drag profile of G1 ballistic coefficient models and modified it to generate an asymmetric surface tension gradient by incorporating a posterior cavity (Supplementary Fig. 3). Following the described design, we fabricated protein motors with characteristic length scales $l$ ranging from 10 mm down to 100 μm (Fig. 1b).

After evaporating the solvent, 20% (w/w) of residual HFIP fuel remains trapped in the protein matrix (Supplementary Fig. 4), hindering the formation of β-sheet nanostructures (Fig. 1c). In the presence of water, the trapped HFIP molecules are replaced by the absorbed water molecules, enabling the self-assembly into β-sheet nanocrystals (as revealed by infrared spectroscopy in Supplementary Fig. 5). When the motors are placed on the water surface, HFIP is slowly released at the air–water interface, creating a surface tension gradient that generates Marangoni forces and propels the protein motor at high speeds. HFIP is very advantageous over other chemical fuels due to its low surface tension, especially at very low concentrations (Supplementary Figs. 6 and 7). Small amounts of HFIP fuel can generate large surface tension gradients, and therefore larger Marangoni propulsive forces are generated compared to other chemical fuels. Furthermore, the entrapment of HFIP in the protein matrix and its slow release through a β-sheet nanocrystal network allow for high storage of fuel and longer mobility lifetimes compared to those of other fuels (Supplementary Table 1). Most organic solvents do not dissolve or even swell the protein, and therefore the amount of other fuel absorbed by the protein matrix is very low (resulting in very short mobility lifetimes in the order of a few seconds). The proposed self-propulsion mechanism is verified against control experiments in HFIP medium, where no locomotion is observed (no surface tension gradient and therefore no Marangoni flow), and against other protein materials (Supplementary Figs. 8 and 9, Supplementary Movie 1). This motor-fuel synergy in our protein-based system, together with the

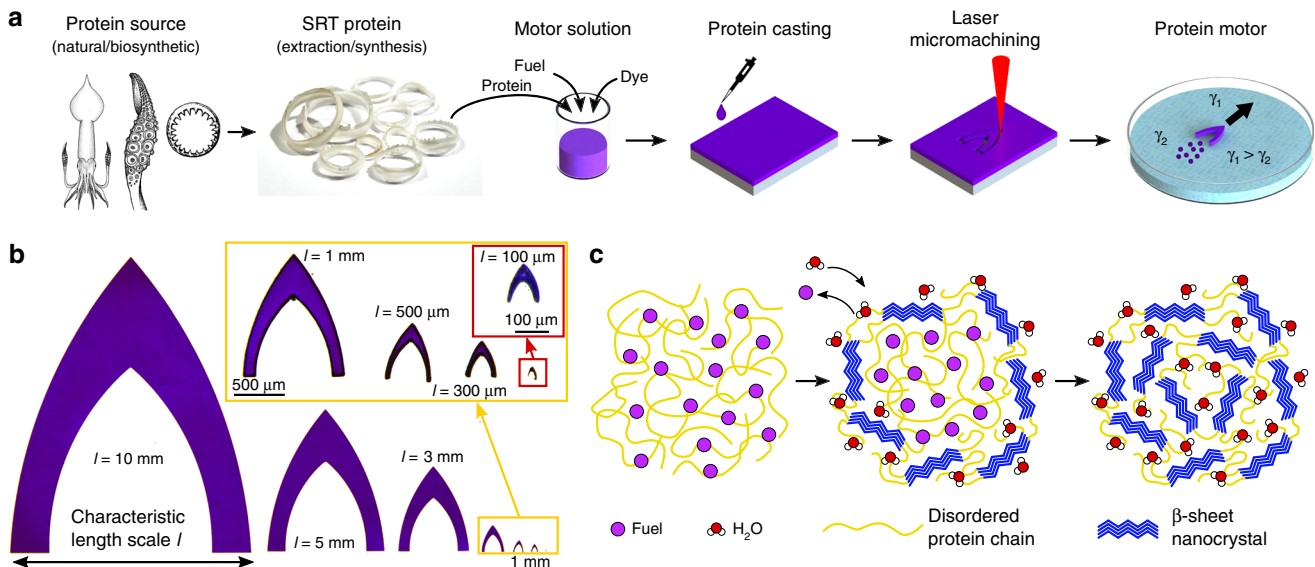

**Fig. 1** Self-propelled protein motors. **a** Fabrication of protein motors: production of the protein either from natural or biosynthetic sources (adapted from reference with permission[54]), SRT extraction and purification, preparation of the motor solution (with SRT protein, fuel, and dyes), protein film casting, laser micromachining of the motors to the specified design shape, and self-propulsion by Marangoni forces at the air–liquid interface. **b** Fabricated protein motor top-view images with characteristic length scales from 10 mm down to 100 μm. **c** Propulsion mechanism: HFIP fuel is initially trapped in the protein matrix, and it is released to the swimming medium when the motor comes in contact with water. Trapped HFIP molecules are replaced by water, inducing the formation of β-sheet nanostructures

miniaturization of the motors (reduction of drag forces), result in superior performance of the motors (i.e., high speeds, long mobility lifetime, and high efficiency). Furthermore, the self-propulsion of protein motors in water is a major advantage over other chemical motor systems that require acidic or basic media, hydrogen peroxide, or chemical reactions for locomotion[28], hence enabling the use of protein-based motors in biological and physiologically relevant environments.

**Propulsion modes of protein motors**. Locomotion and performance of the protein motors was analyzed using deionized (DI) water as the swimming medium. The motors exhibited three propulsion modes depending on their length scale (Supplementary Fig. 10, Supplementary Movie 2): orbiting motion (lateral propulsion, Fig. 2a), straight linear motion (forward propulsion, Fig. 2b), and a combination of the two (Fig. 2c). At large length scales, the surface tension gradient generated by the motor should be isotropic (the posterior cavity is too big for a local increase in fuel concentration), and the motor is laterally propelled in circular orbits (propulsion normal to its longest dimension)[11,18]. At small length scales, however, the motor posterior cavity generates a local increase in concentration of released HFIP at the tail, resulting in an anisotropic surface tension gradient (lower surface tension at the tail) and forward propulsion. We simulated the fuel release across length scales for a quantitative analysis and prediction of the propulsion modes (Supplementary Fig. 11). By analyzing the fuel concentration profile along the posterior cavity and perpendicular to the posterior legs, we introduced a dimensionless parameter $\delta_{propulsion}$ to predict forward vs. lateral propulsion (Supplementary Fig. 12). Large motors ($l = 10$ mm, $\delta_{propulsion} = 1$) have isotropic concentration gradients (i.e., orbiting lateral propulsion dominates), while small motors ($l = 1$ mm and smaller, $\delta_{propulsion} > 1.25$) have anisotropic concentration gradients with a local maximum in the posterior cavity (i.e., forward propulsion is dominant). For intermediate length scales ($l = 3$–$5$ mm, $1 < \delta_{propulsion} < 1.25$), the anisotropy in the concentration gradient around the motor is not strong enough to

guarantee continuous propulsion in the forward direction, and the motor alternates between forward and lateral propulsions. The prediction of propulsion modes with $\delta_{propulsion}$ parameter agrees with our experimental observations, and therefore it is a valid design parameter for programmable propulsion of our protein self-propelled motors.

**Protein motor control**. Locomotion control of most self-propelled systems is challenging due to their lack of directionality, causing uncontrolled motion in random directions. As discussed, the protein motors presented in this work can be designed with preprogrammed propulsion modes, resulting in predefined locomotion with preferred directionality. Taking advantage of such directional propulsion, we demonstrate active direction and trajectory control by magnetic steering (Fig. 3, Supplementary Movie 3). Actuation, propulsion, and steering using external magnetic fields have been previously reported on various microrobots and materials[10,13,29–31]. Here, we fabricated magnetic protein nanocomposite motors by integrating superparamagnetic iron oxide nanoparticles (SPIONs) into the protein matrix. Due to the anisotropic design, the nanocomposite motors align themselves with the direction of an external magnetic field (10 mT), providing active steering control. Therefore, the motors are self-propelled in the forward direction by Marangoni forces while they are steered by magnetic fields. Control experiments show no propulsion but only steering when no chemical fuel is used, validating our hybrid propulsion/steering approach. Using a custom-built electromagnetic coil setup[32], we preprogrammed the variable external magnetic field and steered the motor in complex 2D motion trajectories (a triangle, square, pentagram, and lemniscate) over short time scales. This approach allows for the programmable navigation of protein-based magnetic motors in complex environments, such as path planning.

**Performance of protein motors**. The motors accelerate as they encounter the water surface, instantly achieving a peak velocity

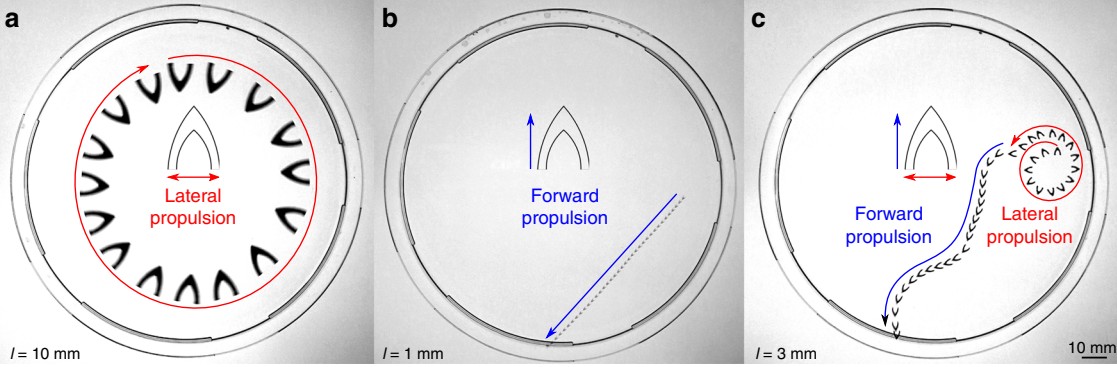

**Fig. 2** Protein motor propulsion modes. Motors with characteristic length scales $l = 10$ mm exhibit lateral propulsion, $l = 1$ mm–100 μm forward propulsion, and $l = 3$–5 mm alternating propulsion between the two modes

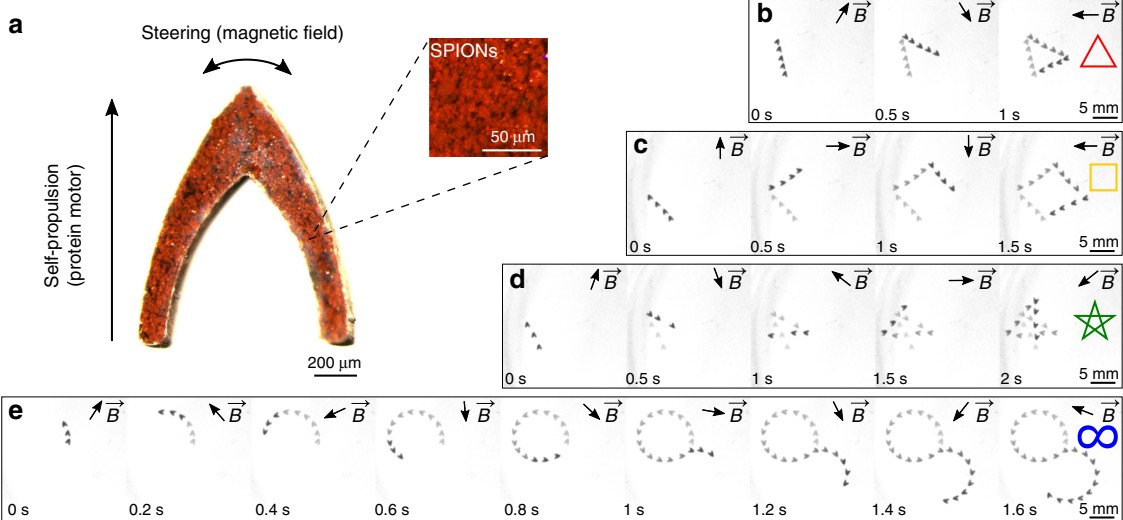

**Fig. 3** Locomotion trajectory control of protein motors ($l = 1$ mm) via active magnetic steering. **a** Magnetic motors (SPION and protein nanocomposite) are self-propelled by the protein motor and steered by a magnetic field. Complex 2D motion trajectories: (**b**) triangle, (**c**) square, (**d**) pentagram, and (**e**) lemniscate trajectories were programmed and defined by a variable magnetic field $\vec{B}$. Arrows show the direction of the magnetic field $\vec{B}$

$v_{max}$ and slowly decreasing velocity during their mobility lifetime. Initially, the motors propel in a continuous regime as the fuel is slowly released from the protein matrix. Afterwards, the fuel at the body edges is partly exhausted and the motors undergo intermittent motions caused by the replenishment of fuel from the body core to the edges[18]. The mobility lifetime of the motors scales with $\sim l$, with an uninterrupted continuous locomotion of 21 min and a total lifetime (scaling with $\sim l^2$) of over 2 h for $l = 10$ mm motors (Fig. 4a, Supplementary Fig. 13). We propose a model, detailed in Supplementary Note 1, to validate our experimental results and to investigate the scaling laws in the protein motor design and performance. The model calculates terminal velocity $v_{max}$ from the Marangoni propulsive forces $F_{prop}$ and drag forces $F_{drag}$ acting on the motor. $F_{prop}$ is approximated to $\sim \Delta \gamma l_{contact}$, where $\Delta \gamma$ is an asymmetric surface tension gradient and $l_{contact}$ is the motor body contact line. To calculate the surface tension gradient, we modeled the fuel release as diffusion from a moving source and experimentally determined the surface tension concentration dependence ($k_{HFIP}$) and diffusion flux from the motor ($J_{SRT}$). We obtain

$$F_{prop} = \frac{2J_{SRT}k_{HFIP}l}{v_{max}} \qquad (1)$$

The drag force scales linearly with velocity (viscous drag, $\sim v$) at low Reynolds number ($Re \ll 1$), and scales quadratically with velocity (inertial drag, $\sim v^2$) at high Reynolds number ($Re \gg 1$). The Reynolds number for the protein motors fall between $Re \approx$ 50–2000 (Supplementary Fig. 14, Supplementary Table 2). Therefore, we cannot ignore inertial effects, and

$$F_{drag} = \frac{1}{2}\rho v_{max}^2 C_d A \qquad (2)$$

where $\rho$ is the fluid density, $v_{max}$ is the velocity, $C_d$ is the drag coefficient, and $A$ is the area. If we equate $F_{prop} - F_{drag} = 0$ at the terminal velocity (no acceleration), we can calculate $v_{max}$ as a function of the motor characteristic length scale design parameter $l$:

$$v_{max} = \sqrt[3]{\frac{8J_{SRT}k_{HFIP}}{\rho C_d l}} \qquad (3)$$

The model is validated using the experimental results (Supplementary Table 3), showing satisfactory agreement with experimentally measured speed of protein motors across length scales. The maximum motor speed scales with $\sim l^{1/3}$, increasing with miniaturization of the motors, and reaching peak velocities up to 408 mm s$^{-1}$ for $l = 100$ μm motors (Fig. 4a).

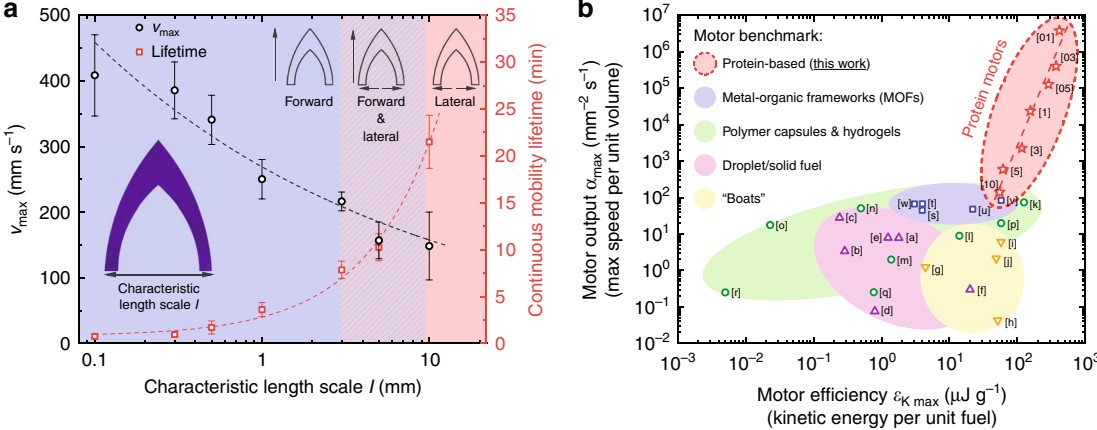

**Fig. 4** Protein motor performance. **a** Scaling of motion speed and lifetime. Maximum speed of the motors scales with $\sim l^{-1/3}$ as described by the model (dashed line), up to a maximum speed of 408 mm s$^{-1}$ for $l = 100$ µm motors. Continuous mobility scales linearly with $l$ up to 21 min for $l = 10$ mm motors and a total lifetime of over 2 h. Black circles are $v_{max}$, red squares are continuous mobility lifetime. Error bars represent standard deviation, $n = 15$. **b** Motor benchmark. Significant metrics for comparing performance of chemical motors are motor performance output $\alpha_{max}$ (maximum speed per unit volume) and motor efficiency $\varepsilon_{K\ max}$ (maximum kinetic energy per unit fuel). Protein motors outperform other chemical Marangoni motors from a diversity of materials and fuels. Red stars are protein-based motors (this work), blue squares are MOF motors, green circles are polymer motors, purple triangles are droplet/ solid fuel motors, yellow inverted triangles are boat motors. Full references are given in Supplementary Table 4

Since motors carry a finite amount of fuel, quantifying the motor efficiency is necessary to evaluate the performance of our protein motors and compare it to previous reports on analogous motor systems. Typical benchmarking of chemical motors uses metrics, such as motor performance output $\alpha_{max} = v_{max}/V_{motor}$ (maximum speed per unit volume) and motor efficiency $\varepsilon_{K\ max} = m_{motor}v_{max}^2/(2m_{fuel})$ (maximum kinetic energy per unit mass of fuel)[18]. In these terms, our protein motors compare very favorably to other surface tension-driven chemical motors, including metal-organic frameworks and polymeric systems, by several orders of magnitude in both $\alpha_{max}$ and $\varepsilon_{K\ max}$ (Fig. 4b, Supplementary Table 4). We attribute such superior metrics and performance to the combination of: (a) low-surface tension of HFIP fuel (large Marangoni propulsive forces with small amount of fuel), (b) entrapment of fuel inside the protein matrix (slow release of fuel and long mobility lifetime), and (c) the miniaturization of the motors (motor design and fabrication, and reduction of drag forces acting on the motor).

**On-demand termination and degradation**. Although high speeds and long mobility lifetimes are desirable for the performance of the motor, they might be a disadvantage for applications where the motor is required to reach a specific destination and remain there to perform its function (e.g., targeted cargo delivery). In this section, we investigate on-demand termination and self-destruction of the protein motors via different environmental stimuli (Fig. 5a). Due to the dynamic nanostructure of SRT proteins (stabilized by reversible β-sheet nanocrystals), the protein matrix can be altered by stimuli such as pH, biomolecule concentration gradients (urea, enzymes, etc.), and temperature. If the β-sheet nanocrystals are broken, the cross-linking structures in the protein network will be removed and the protein matrix will be ultimately degraded. In such case, the protein matrix cannot store the fuel in its broken network and will release its entirety to the swimming medium (effectively terminating the locomotion). SRT protein charge is dominated by histidine amino acids (11%), and therefore has an isoelectric point of ~6.7, positive charge at lower pH and negative charge at higher pH[33]. At low and high pH, electrostatic repulsive forces disrupt the formation of β-sheets, opening the protein network and terminating

the locomotion (Fig. 5b, Supplementary Fig. 16). A similar phenomenon occurs when operating the protein motors in the presence of urea, which disrupts the hydrogen bonding of β-sheets, yielding a soft, non-crosslinked, and entangled protein network[34] (Fig. 5c, Supplementary Fig. 17). With increasing concentration of urea, β-sheet nanostructures are disassembled (measured by infrared spectroscopy in Supplementary Fig. 18), facilitating the release of fuel through a non-cross-linked protein network (reducing the motor mobility lifetime). On the other hand, temperature does not disrupt the β-sheet nanostructures (at least, not in the 5–70 °C range), but affects the amorphous chains instead. Hydrated native SRT proteins have a glass transition temperature $T_g$ around 35 °C between a rigid, glassy protein network and a soft, flexible network[35]. As temperature increases past the $T_g$, the previously trapped fuel is released through the loose network, reducing the mobility lifetime (Fig. 5d, Supplementary Fig. 19). Hence, the locomotion of protein motors can be controlled by local stimuli causing a response in the protein nanostructure, enabling on-demand termination (off switch).

In addition to on-demand termination of the motors, we investigate their self-destruction via degradation of the protein matrix. Protein motors are naturally degraded by the disruption of the hydrogen bonding network or selective enzymatic degradation (proteolysis), which can be exploited for active cargo delivery applications[35–37]. We demonstrated such capability by administering a local pH stimulus (diluted yellow-dyed acetic acid) to a free-swimming protein motor ($l = 10$ mm) with a violet dye as a drug model (Fig. 5e, Supplementary Movie 4). As the β-sheet crosslinks are disrupted by the acidic environment, the protein motor moves towards the pH stimulus (pH-taxis)[17,38], and progressively swells and degrades, releasing its cargo (dye) into the swimming medium. This cargo delivery approach is extendable to multiple drugs due to the facile fabrication and encapsulation processes. As a proof of concept, we demonstrated the release of doxorubicin (DOX, a commonly used fluorescent chemotherapy agent) on stimuli-responsive protein motors with characteristic length scales $l = 500$ µm and $l = 100$ µm (Fig. 5f, Supplementary Fig. 20). A pH stimulus (diluted acetic acid drop) was placed on top of the DOX-loaded motors. The motors in contact with the stimulus degraded over time releasing the

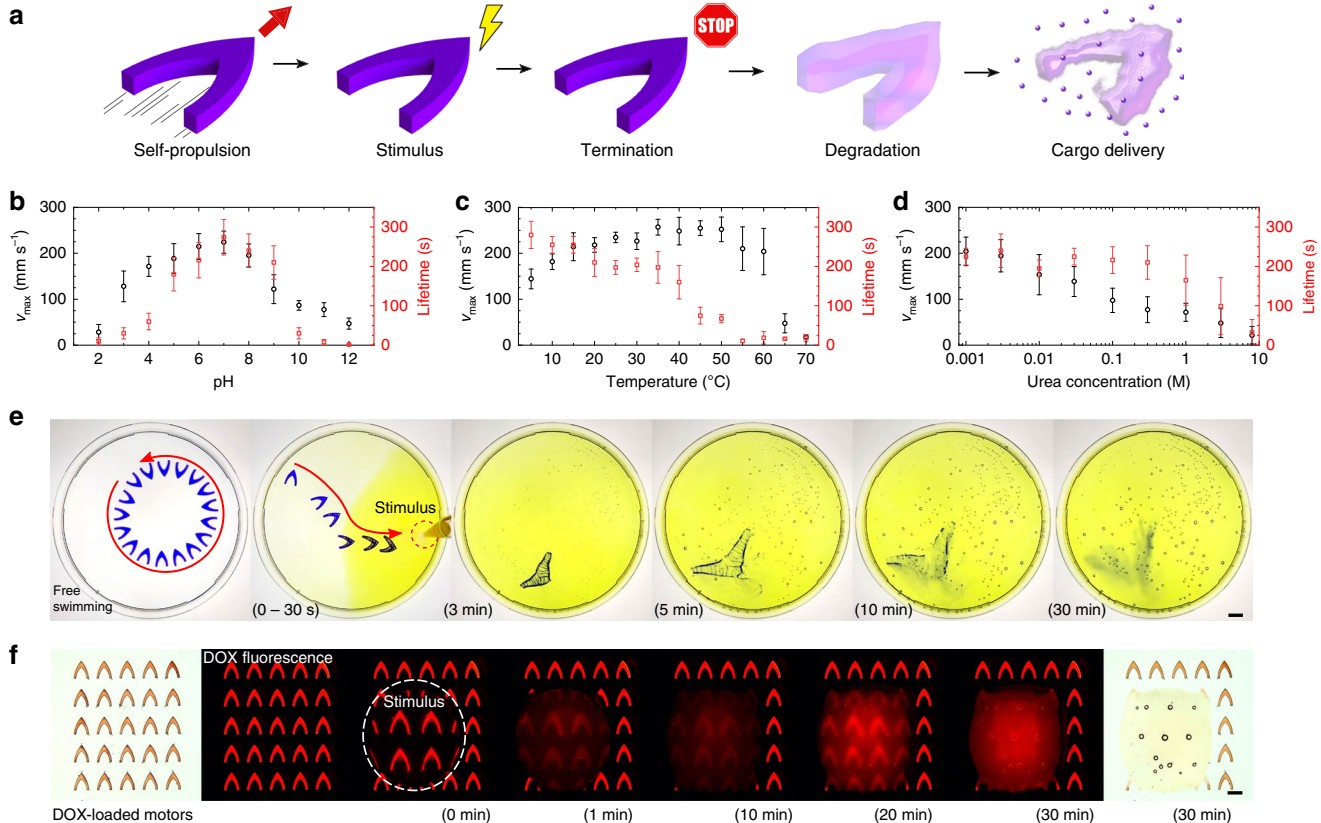

**Fig. 5 a** Protein motor on-demand termination and degradation for cargo delivery. **b** pH, **c** urea, and **d** temperature stimuli reduce the mobility lifetime and ultimately terminate the locomotion. Black circles are $v_{max}$, red squares are continuous mobility lifetime. Error bars represent standard deviation, n = 15. **e** Degradation and cargo release. Free-swimming $l$ = 10 mm motor reacts to a pH stimulus by moving towards low pH and reducing its lifetime. Once stopped, the motor swells and degrades over 30 min, releasing its encapsulated cargo (scale bar: 10 mm). **f** Stimuli-responsive release of DOX from $l$ = 500 μm motor (scale bar: 500 μm)

encapsulated DOX (increasing fluorescence intensity in the droplet). On the other hand, motors in $H_2O$ (no pH stimulus) were stable and retained the encapsulated DOX, demonstrating stimuli-responsive degradation and release.

Most synthetic chemical motors present biocompatibility challenges because: (i) the fuel is highly cytotoxic, (ii) the swimming media requires hazardous chemicals, or (iii) the motor itself is not biocompatible. However, the protein motors described in this work present a fully biocompatible alternative. The fuel, HFIP, is a metabolite of sevoflurane (FDA-approved and widely used inhalation general anesthetic, Supplementary Note 2)[24,25]. Hazardous chemicals are not required in the liquid media, thus the protein motors can operate in water and other physiological fluids. The protein that constitutes the motor is not only biocompatible, as previously demonstrated in in vitro and in vivo studies[39,40], but also biodegradable. The biocompatibility and biodegradability, together with the versatile cargo delivery via self-destruction strategy, make the protein motors attractive for future minimally invasive medical operations in physiological environments with an air–liquid or immiscible liquid interfaces (either naturally occurring interfaces like in stomach or lung alveoli[41–43], or engineered interfaces by integration of air cavities and bubbles).

**Multifunctional protein motors**. Although various chemical surface motors have been reported to date, the majority of the motors are non-functional passive elements used solely for fuel storage purposes with little control over the locomotion. Here, in addition to cargo delivery via degradation, we explore diverse functionalities of the protein motors. First, we propose the use of SRT proteins as integrated modular motors (Supplementary Movie 5). We powered inanimate objects with programmable locomotion by selectively coating the propulsive regions with fuel-loaded protein. Shark fin swimmers (Fig. 6a) exhibit continuous forward locomotion according to the position of the integrated modular motor. This approach was also applied to power autonomous mechanisms, such as gear trains (Fig. 6b) by selectively coating the gear teeth. We demonstrate successful power transmission in speed reducer and multiplier gear trains for at least 40 min. Depending on the position of the integrated modular motor, rotors spin clockwise or counterclockwise (Fig. 6c). We also demonstrated the use of protein motors as integrated modular power sources in self-propelled magnetic microrobots (Fig. 6d, Supplementary Fig. 21). Synthetic magnetic microrobots were 3D-printed by two-photon lithography and coated with a 100 nm-thick cobalt magnetic layer[32]. Next, the modular protein motor was integrated by selectively coating the propulsive area of the microrobot. Microrobot was steered on the water surface by an external magnetic field (5 mT) while being propelled by the integrated protein motor (Marangoni forces), enabling the programming of complex locomotion over short time scales.

Exploring additional functionalities, we examined the use of SRT protein motors for environmental remediation (Fig. 6e, Supplementary Fig. 22). Industrial activities such as mining, processing of ore, fabrication of electronic devices, and use of

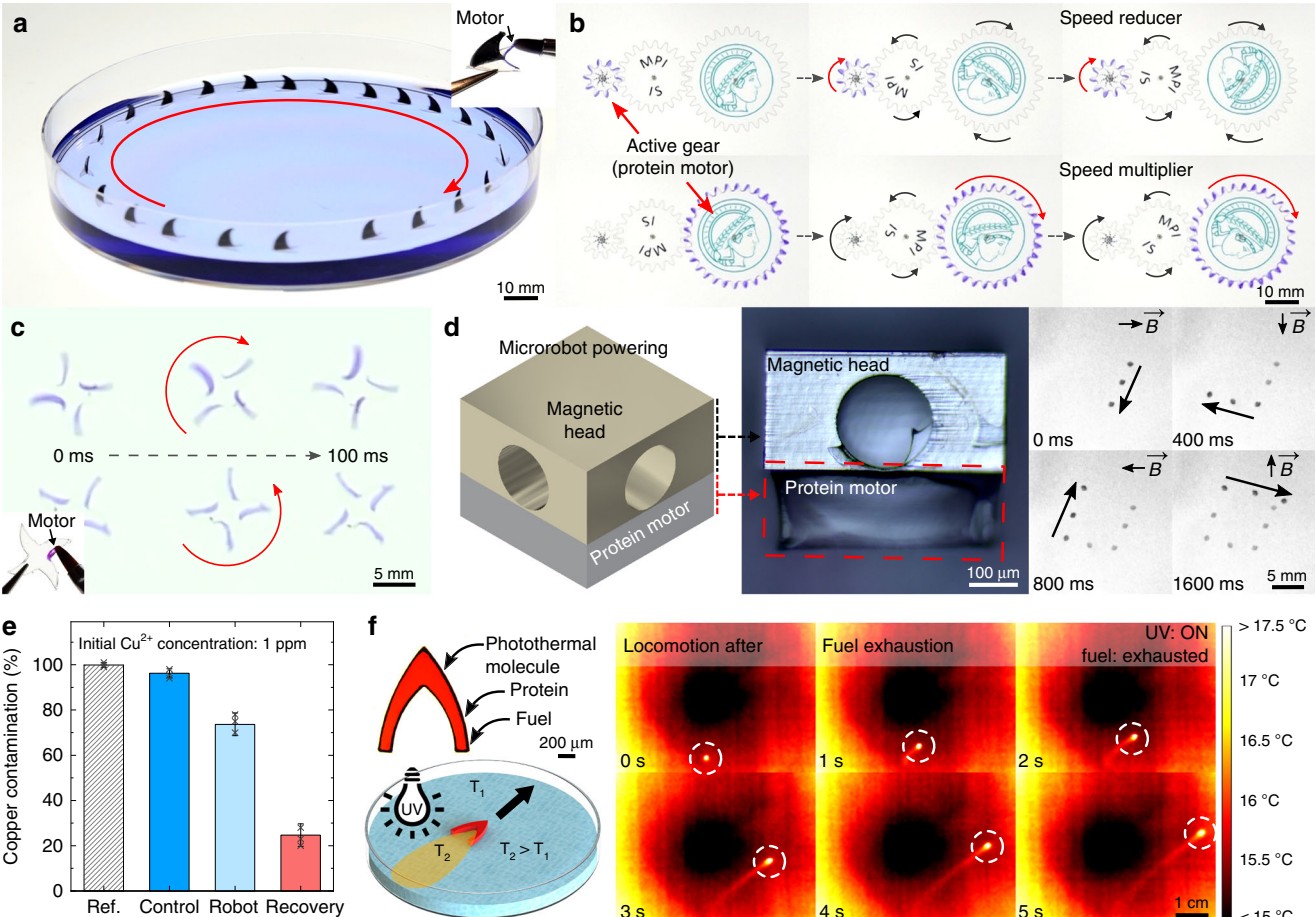

**Fig. 6** Multifunctional protein motors. Modular motors: Protein coatings are applied to inanimate objects as integrated power sources for locomotion: **a** shark fin swimmers, **b** autonomous gear trains (reducer and multiplier), **c** spinning rotors, and **d** magnetic microrobots with programmable complex locomotion (arrows show the direction of the magnetic field $\vec{B}$). **e** Environmental remediation. Single protein motor ($l = 10$ mm) captures $Cu^{2+}$ contaminants from water. Metal ions can be recovered after treatment (error bars represent standard deviation, $n = 3$). **f** Locomotion after fuel exhaustion. Protein motors ($l = 1$ mm) were doped with photothermal molecules. After the chemical fuel is exhausted, UV wide-field illumination induces thermal Marangoni forces that propel the motor forward. Color scale represents temperature from 15 to 17.5 °C

pesticides have led to the release of heavy metal contaminants into industrial wastewaters, including copper ions. As a result, high concentration of pollutants has not only been found in hydrological resources, but also in soil, fish, and crops, in levels that are not safe for human activity or consumption[44]. Therefore, heavy metal water pollution (including copper ion contamination) is an increasing global concern and poses an immediate threat to human and animal life (both terrestrial and aquatic). Among the technologies that are being developed to fight this environmental problem, micromotors and nanomotors are rising as promising platforms for efficient removal of organic and inorganic contaminants[28,45,46]. Micromotors for heavy metal removal usually are composed of adsorptive materials (typically carbon-based nanomaterials)[47] or are functionalized with chelating agents (molecules that bind to metal ions)[48]. In the case of our protein-based micromotors, we took advantage of the natural amino acid histidine, present in SRT proteins (~10 mol%). Histidine are known natural chelators that bind to copper ions (among others), and are commonly used in protein purification processes (histidine-tag sequence modification)[49]. Although the remediation efficiency could be further improved (e.g., by decreasing the motor size and using multiple motors simultaneously), these results demonstrate the metal ion sequestration

capabilities of our protein motors for potential future environmental remediation applications, and add multifunctionality to our versatile, non-toxic, and self-propelled platform.

An intrinsic limitation of self-propelled motors is the availability of chemical fuel, since motors have no means of doing work when the fuel is completely exhausted. Previous reports explored the possibility of refueling after fuel exhaustion, but it requires retrieving the motor from the swimming medium, drying, refueling, and transferring again to the swimming medium[18]. Multiple refueling cycles can degrade the motors and deteriorate their performance[18]. In this work, we induced locomotion to the protein motors after the fuel was exhausted via non-contact photothermal propulsion, giving the motors a second lifetime (in addition to their chemical fuel-driven mobility lifetime). Light-induced small-scale manipulation via thermocapillary effect has been previously demonstrated by locally heating up the liquid medium, however, it usually requires fine positioning and focusing in order to control the particle trajectory[50,51]. Here, we integrated active molecules (Disperse Red 1 dye) into the protein motor and photothermally induced Marangoni propulsion by wide-field UV illumination (Fig. 6f, Supplementary Movie 6). Disperse Red 1 is an azobenzene dye that undergoes reversible photoisomerization, converting light

into thermal energy by releasing heat into the encapsulating matrix[52]. Due to the fast photoisomerization of Disperse Red 1 and the high thermal conductivity of SRT proteins[53], the motors were quickly and homogeneously heated upon wide-field UV illumination. When the motor was heated, the temperature of the surrounding swimming fluid increased, self-generating a temperature gradient along the contact line. Due to the specific geometry of the motor design, a local temperature maximum in the fluid at the posterior part of the motor created an anisotropic temperature gradient (Supplementary Fig. 23). Since the surface temperature of water decreases with temperature, photothermal heating of the motors caused an anisotropic surface tension gradient (with the lowest surface tension in the posterior part of the motor) that generated the Marangoni forward propulsion. Infrared thermal images of the locomotion after exhaustion of the fuel showed that only the motor was heated from wide-field illumination while the liquid pool was unaffected, demonstrating that long-range forward locomotion (without chemical fuel) originated from the design of the protein motor. This method offers a non-contact, continuous transition between chemically driven and thermally driven propulsion modes (giving the motors a second operational lifetime).

## Discussion

We have developed fast, efficient, and degradable self-propelled motors using a squid-derived protein as the stimuli-responsive and biodegradable motor matrix and a metabolite of inhalation anesthetics as the fuel. The tunable nanostructure of the protein matrix regulates the release of trapped fuel, generating anisotropic surface tension gradients, and Marangoni forces that self-propel the motors. This protein motor system overcomes major limitations in the field, which are the lack of directionality and control, poor performance (efficiency, speed, and mobility), and toxicity of materials (motor, fuel, or swimming medium). The protein motors presented in this work surpass previous Marangoni motors in terms of performance output and efficiency by several orders of magnitude, making them the best performing liquid surface motors reported to date. We describe control strategies to regulate the propulsion modes of the motors (by motor design), their speed and lifetime (including on-demand termination by local stimuli), and their directionality (active steering by external stimuli), resulting in programmable complex locomotion and trajectories. Furthermore, we demonstrate diverse functionalities of the protein motors in environmental remediation applications (removal of water contaminants), microrobot powering (powering of inanimate objects), and cargo delivery via self-destruction and degradation for targeted release applications. This protein motor can be integrated onto virtually any material as a modular propulsion source, and can be functionalized with diverse nanoparticles and biomolecules, opening up the design space for control, sensing, and actuation schemes for small-scale robots, machines, and devices. To our knowledge, this is the first biodegradable self-propelled surface motor that is capable of such multifunctionality, high performance, and precise control at the air–liquid interface. Future work will explore the protein motors as modular biodegradable propulsion sources in microrobotics for minimally invasive medical operations in physiological environments with natural or engineered air–liquid interfaces for sensing and therapeutic applications.

## Methods

**Motor fabrication**. SRT were extracted from the tentacles of *L. vulgaris* squid from Tarragona (Spain)[35]. SRT protein was dissolved in HFIP to a concentration of 50 mg mL$^{-1}$, and 1% crystal violet dye was added. 100 μL of solution were cast on polydimethylsiloxane substrates and left to evaporate for at least 3 h, yielding 20 μm protein films. The films were cut in a LPKF ProtoLaser U3 laser cutter (0.189 W, 50 kHz) to the desired motor design (characteristic lengths from 100 μm to 10 mm, Supplementary Fig. 3). Laser micromachining details are described in Supplementary Fig. 2. After machining, an array of motors were mechanically peeled off from the substrate (tweezers and needle tip) and transferred for characterization and analysis. The motors, cut from the same protein film, had homogeneous chemical composition (i.e., equal fuel storage) after the fabrication process (Supplementary Note 3). The motors were imaged in a Leica M205 FA stereomicroscope and in a Keyence VK-X200 3D laser scanning microscope.

**Protein characterization**. Protein nanostructure and fuel were analyzed by Fourier transform infrared spectroscopy (FTIR) in a Bruker Tensor 2 with a Platinum ATR accessory (256 scans with 4 cm$^{-1}$ resolution). Fuel content in protein motors was measured by thermogravimetric analysis (TGA) in a Netzsch STA 449 F3 Jupiter coupled to a Bruker Alpha FTIR spectrometer at 10 °C min$^{-1}$ under 90 mL min$^{-1}$ argon flow. The mechanical properties of protein films as function of temperature were measured by oscillatory rheology in a TA Discovery HR-2 rheometer.

**Motor performance analysis**. The motors were transferred to Petri dishes with DI water and their locomotion was recorded. Motors with $l = 1$–$10$ mm were analyzed in ø140 mm Petri dishes and recorded with a Sony HDR-CX900 camera (25 fps). Motors with $l = 100$–$500$ μm were analyzed in ø35 mm Petri dishes and recorded with a Basler acA2440-75 μm camera (70 fps) mounted on a Zeiss Stemi 305 stereomicroscope. Trajectories and velocities were analyzed in Fiji software. In motor control experiments, solutions of pH 2–12, urea concentrations of 1 mM–8 M, and temperatures from 5 to 70 °C were used.

**Magnetic motor**. Magnetic nanocomposite motors were fabricated by dispersing (vortexing) <50 nm SPIONs in 50 mg mL$^{-1}$ (SRT/HFIP) to a 1:3 ratio ($w_{SPION}/w_{protein}$). The resulting solution was cast and micromachined as previously described (analogous to non-magnetic motors).

**Magnetic steering**. A custom eight-electromagnetic coil system was used to generate variable magnetic fields to steer magnetized protein motors[32]. The electric currents through the coils were calculated to minimize the magnetic field spatial gradient, which would cause translational force, at the center of the workspace. Fuel-depleted magnetic protein motors did not exhibit translational motion on the water surface under the influence of the magnetic field sequences.

**Degradation tests**. Dyed (crystal violet) motors ($l = 10$ mm) were placed on ø140 mm Petri dishes with DI water, swimming freely. Then, a pH stimulus (acetic acid with yellow food colorant) was administered on the edge of the Petri dish to a total concentration of 10% acetic acid. The swimming, swelling, and dye release was recorded with a Sony HDR-CX900 camera (25 fps). 5% DOX (50 μg$_{DOX}$ mg$_{protein}^{-1}$) was incorporated to the motors solution and motors were fabricated as previously described. DOX-loaded motors were degraded with 10 μL droplets of 5% diluted acetic acid.

**Modular motor experiments**. Polyethylene terephthalate (PET) films (100 μm thick) were micromachined into rotors, shark fin swimmers, and m1 gears (z10, z20, and z30). Dyed 50 mg mL$^{-1}$ SRT/HFIP solution was applied to the desired propulsion areas with a small paintbrush, and it was left to dry for 1 h. Synthetic magnetic microrobots were 3D-printed by two-photon lithography and coated with a 100 nm-thick cobalt magnetic film. The microrobots were inserted in polyurethane rubber molds, and 1 μL of 50 mg mL$^{-1}$ SRT/HFIP solution was cast on the microrobot side (Supplementary Fig. 21). Then, the coated objects were transferred to pools of DI water for testing.

**Environmental remediation experiments**. Copper (II) nitrate hemipentahydrate was dissolved to a concentration of 1 ppm Cu$^{2+}$ as reference. One single motor ($l = 10$ mm) was placed in 5 mL 1 ppm solution for 1 h while stirring the solution with a small magnetic bar. After this, the motor was dissolved in 5 mL 10% acetic acid. The Cu$^{2+}$ content of the solutions was measured by inductively coupled plasma-optical emission spectrometry (ICP-OES). Cu$^{2+}$ concentration was measured at 324.754 nm wavelength based on 0.02–15 ppm calibration curves.

**Photothermal motor experiments**. 1% disperse red photothermal dye was added to the 50 mg mL$^{-1}$ SRT/HFIP solution. The solution was cast and micromachined into the motor design as previously described. The motors were fully immersed in DI water for 2 h to remove the HFIP (i.e., no fuel). Then, the motors were illuminated with an Omnicure series 2000 UV lamp (broadband 320–500 nm). The temperature of the photothermal motor was monitored with a FLIR ETS320 infrared camera.

## Data availability

The authors declare that data supporting the findings of this study are available within the paper and its supplementary information files.

## Code availability

This study did not use custom code that is deemed central to the conclusions.

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

## Acknowledgements

This project was funded by the Max Planck Society. A.P.-F. thanks the Alexander von Humboldt Foundation for the Humboldt Postdoctoral Research Fellowship, and the Federal Ministry for Education and Research. We thank Cora Bubeck, Samir Hammoud, and Hamed Shahsavan for help with TGA, ICP, and photothermal experiments, respectively.

## Author contributions

A.P.-F. and M.S. conceived and coordinated the project. A.P.-F. performed the experiments, analyzed the data, and wrote the manuscript. J.G. performed the magnetic steering experiments and COMSOL simulations. All authors participated in manuscript revisions, discussion, and interpretation of the data.

## Additional information

**Competing interests:** The authors declare no competing interests.

