## [Peer Review File · Nature Communications]

Reviewers' comments:

Reviewer #1 (Remarks to the Author):

The authors report the development of a very unique motor by depositing the beta-sheets of protein nanocrystals on substrates. They cut the motor into unique pieces using laser micromachining and then once immerse inside deionized water, the motors self-propel owing to the surface tension gradient at the interface. They also show motions due to zeta-potential and thermal gradients on the same embodiment with some minor modifications. The novelty of the work lies in the development of a biocompatible motor which can translate in water medium with multiple stimulants. The speed reported in the study are also found to be exceptionally high, as compared to the previously reported ones. The study also showcases a number of complex motions possible using the same motor. Even if I liked the work very much but feel this is perhaps suitable to more specialized journal for the following reasons,

(i) The concepts reported such as surface tension, zeta-potential or thermal gradient driven motions are not reported for the first time. The novelty lies only in the material usage. Felt that the authors should have deliberated more on the origin of the motors to bring in some more novelty.

(ii) The authors even do not attempt to elucidate the reasons for such high speeds with some calculations. The consequences of the same (say transition from laminar to weakly turbulent regimes) have also not been discussed. Thus, the scientific aspects appear very weak while the demonstrations remain to be the strongest part of the work. One may ask the question that why suddenly the use of protein molecules may create such a high surface tension gradient which may lead to such high speed propulsion? Or it has some other origin? In that case, the proposed mechanism may be challenged.

(iii) Grossly the paper is written in such a manner where the previous works are not very well respected. In fact, many of the seminal contributions of the previous works have been ignored. Further, the paper can also be criticized for heavy self-citation.

(iv) What type of environmental remediation is expected with these motors? Any example case has not been shown here. The size of the motor excludes its potential in the usage of drug delivery and all. The motor may migrate only at the interface and that also for a small amount of time until it releases the protein to create the surface tension gradient. Perhaps a magnetic handle would have been ideal for such a motor for recovery and recharge.

Thus, although I liked the paper much, could not recommend it for Nature Communications. Thank you very much for putting up such a beautiful piece of work.

Reviewer #2 (Remarks to the Author):

This paper reported a versatile protein motor made of squid ring teeth protein. The authors discussed in good details the motor propulsion mechanism, the performance, methods to control the properties of the motor, and several interesting alternative implementations and application-relevant studies in environmental cleaning and self-destructing cargo delivery.

The work is generally very well done, the performance of the motor is very impressive. This is a very solid progress in surface tension driven mini-micro robotic agents. The paper also demonstrates the potential applicability, which may interest researchers in related fields.

One thing that is not clearly discussed in the paper is the lateral motion of the large version of the motor, and the chaotic motion of the smaller version. In theory, the symmetric shape should lead to no motion in lateral direction. So the motion must come from the nonuniformity of the shape/protein concentration etc. The authors should discuss this at least qualitatively. If possible, the uniformity of the protein concentration of the materials across the sheet (e.g. of Fig. 1c) could be quantitatively analyzed.

For the matter of replicability, the authors could give more details in the fabrication steps of the motor. For example, one interesting step is the peeling off, which may deform the structure at a great

degree and lead to asymmetry?

Reviewer #3 (Remarks to the Author):

This article reports on self-propelled protein-based motors made of squid ring teeth with hexafluoroisopropanol (HFIP) as fuel to generate Marangoni propulsion. The material used is of high interest due to several properties: biocompatibility, biodegradability and also due to the structural variation of the material (cross linking related to beta-sheet networking) that can vary according to the preparation method or to the nature of the environment. The versatility of the material is at the origin of the high interest of the present system for which multiple functionalities have been explored (motility response to the chemical properties of the environment, trajectory control by magnetic steering, environment remediation...). The possibilities and efficiency of the system are clearly demonstrated by well designed experiments and deserves publication although there are several important points that should be further analyzed to reach the standards of Nature Communications publications.

The main question is: why is that system faster and longer lasting than most systems described in the literature? There is finally only one sentence, not bringing much insight, to explain this: We attribute such remarkable metrics to the slow release of the fuel due to entrapment in the protein nanostructure, and to the down scalability (miniaturization) of the motors.

The surface tension properties of the fuel solvent given in the SI should be discussed in regard to solvent commonly used in such systems. The parameters used in the model proposed in SI should be more explicit and used to predict more effects than only the size of the motor. The authors attribute the decrease of the maximum speed at low and high pH, in the presence of urea and as a function of temperature (see below) to a higher release of fuel. If higher quantities of fuel are released on the surface, the expected effect would be an acceleration of the motor and not the contrary (the model used by the authors would also predict an increase of the maximum speed if the fuel diffusion flux is increased). Furthermore, in the case of temperature, the effect after the glass transition is rather an increase of the velocity (Figure 3c)) than a decrease as claimed by the authors.

The possibility to use magnetic field to drive the motor is very interesting but the bibliography proposed for a non-original approach is really poor. I would suggest adding more references on that part as for instance A. Ghosh and P. Fischer, *Nano Lett.* 2009, 9, 2243.

The part dedicated to the photothermally induced motion after fuel is exhausted is unclear. What is the dye used? How are the surface tension gradients generated?

I would also suggest reducing the information provided in each figure. On the printed version they are not readable and some of the information provided could be suppressed or reduced.

Reviewer #1 (Remarks to the Author):

The authors report the development of a very unique motor by depositing the beta-sheets of protein nanocrystals on substrates. They cut the motor into unique pieces using laser micromachining and then once immerse inside deionized water, the motors self-propel owing to the surface tension gradient at the interface. They also show motions due to zeta-potential and thermal gradients on the same embodiment with some minor modifications. The novelty of the work lies in the development of a biocompatible motor which can translate in water medium with multiple stimulants. The speed reported in the study are also found to be exceptionally high, as compared to the previously reported ones. The study also showcases a number of complex motions possible using the same motor.

We thank the reviewer for carefully reading the paper and for providing very helpful comments. We will address the comments on a point-by-point basis below:

Even if I liked the work very much but feel this is perhaps suitable to more specialized journal for the following reasons:

(i) The concepts reported such as surface tension, zeta-potential or thermal gradient driven motions are not reported for the first time. The novelty lies only in the material usage. Felt that the authors should have deliberated more on the origin of the motors to bring in some more novelty.

Indeed, the concepts of surface tension, zeta-potential, and thermal gradient driven motions are not reported for the first time. Such phenomena have been exploited in a great diversity of self-propelled systems, from active microdroplets to milli/centimeter-scale robotic platforms^{1,2}. Such self-propelled systems have significant limitations: low fuel efficiency (they require high volume of fuel), short mobility lifetime, moderate speeds, random locomotion, difficult miniaturization, and material toxicity³. Despite recent research on new highly-absorbent materials for improved fuel storage⁴⁻⁶, these limitations are current roadblocks in the development of self-propelled functional motors (not just uncontrolled, non-functional vehicles).

In this work, we have developed for the first time multifunctional and fully biodegradable self-propelled motors from squid-derived proteins and an anesthetic metabolite. Due to the tunable structure of the proteins and their unique interaction with the fuel, our protein-based motors overcome major limitations in the self-propelled robotics field: excellent performance output and efficiency (orders of magnitude higher than previous reports), high mobility lifetime, miniaturizable fabrication, controlled and programmable locomotion, on-demand degradation at the end of their lifetime, biocompatibility/biodegradability, and multifunctionality (environmental remediation, microrobot powering, and cargo delivery).

We have revised the main text and introduction in order to clarify current limitations in the field and highlight the novelty of our work.

(ii) The authors even do not attempt to elucidate the reasons for such high speeds with some calculations. The consequences of the same (say transition from laminar to weakly turbulent regimes) have also not been discussed. Thus, the scientific aspects appear very weak while the demonstrations remain to be the strongest part of the work.

We respectfully disagree with the reviewer on this argument. We did include a model and calculations accounting for the motor speed in the original submission. However, for a better and more comprehensive description of the physics driving our motors, we have revised the model in the Supplementary Information

(including laminar vs turbulent discussions) and included a summarized version in the main text. We describe the model below:

We propose a propulsion model, which builds on previous work by Ayusman Sen's group⁷, with the purpose of validating our experimental results and investigating the scaling laws in the protein motor design and performance. The model is based on Newton's second law. For a motor moving at terminal velocity (maximum velocity, with no acceleration), the forces acting on the motor are:

$$F_{prop} - F_{drag} = 0 \quad (\text{R1.1})$$

Where F_{prop} is the asymmetric surface tension force along the body contact line, and F_{drag} is the viscous drag force.

Propulsion force

The propulsion force F_{prop} , caused by the release of HFIP along the contact line and the resulting surface tension gradient, is:

$$F_{prop} = \int_{\text{contact line}} \sigma \mathbf{s} dl \sim \Delta\gamma l_{\text{contact}} \quad (\text{R1.2})$$

where σ is the force per length in a direction tangent to the surface, \mathbf{s} is the unit vector tangent to the free surface and normal to the contact line, and dl is an incremental arclength along the contact line. If σ is constant along the contact line, the resulting F_{prop} is zero, but an asymmetric σ profile results in a net Marangoni propulsive force. For simplicity, the propulsive force can be approximated to $\sim \Delta\gamma l_{\text{contact}}$, where $\Delta\gamma$ is an asymmetric surface tension gradient and l_{contact} is the motor body contact line. For our current motor design, $l_{\text{contact}} \approx 2l$, where l is the characteristic length design parameter (see **Figure S3**). Therefore, $F_{prop} \approx 2 \Delta\gamma l$.

The surface tension gradient, $\Delta\gamma = c k_{\text{HFIP}}$, depends on the local concentration of released fuel c and on a calibration parameter k_{HFIP} . The parameter $k_{\text{HFIP}} = 1.39 \cdot 10^{-4} \text{ N m}^2 \text{ mol}^{-1}$ was calculated from HFIP pendant drop surface tension calibration measurements with a linear approximation at low concentrations (**Figure S6**).

The local concentration c can be calculated as the amount of fuel released in a time interval ($J_{\text{SRT}} A \Delta t$) per traveled volume ($v A \Delta t$), where J_{SRT} is the diffusion flux (moles of released fuel per unit area and per second), v is the velocity, A is the area, and Δt is the time interval. $J_{\text{SRT}} = 6.65 \cdot 10^{-1} \text{ mol m}^{-2} \text{ s}^{-1}$ was calculated from **Figure S9** as a linear approximation for very small Δt . Therefore:

$$F_{prop} = \frac{2 J_{\text{SRT}} k_{\text{HFIP}} l}{v} \quad (\text{R1.3})$$

Drag force

The drag force scales linearly with velocity (viscous drag, $\sim v$) at low Reynolds number ($Re \ll 1$), and scales quadratically with velocity (inertial drag, $\sim v^2$) at high Reynolds number ($Re \gg 1$). The Reynolds number for the protein motors fall between $Re \approx 50 - 2000$ (**Table R1.1**). We simulated the flow velocity field around motors with characteristic length l from $100 \mu\text{m}$ to 10 mm in COMSOL with inlet velocity v_0 (maximum speed measured experimentally, **Table R1.1**) (**Figure R1.1**).

Figure R1.1: COMSOL-simulated flow velocity field around motors with characteristic length l from $100 \mu\text{m}$ to 10 mm and with inlet velocity v_0 (maximum speed measured experimentally, **Table R1.1**).

Table R1.1: experimental speed v_0 and Reynolds number for protein motors with characteristic length scale l

l (mm)	v_0 (mm/s)	Re
0.1	408.2	54
0.3	385.3	153
0.5	340.6	226
1	250.3	332
3	216.3	862
5	156.9	1042
10	148.5	1971

We observe that $l = 100 \mu\text{m}$ and $300 \mu\text{m}$ motors exhibit laminar flow, while motors with characteristic length scale $l = 500 \mu\text{m}$ and higher exhibit turbulent flow (with periodic vortices), indicating that our motors

perform in the transition regime between laminar and weakly turbulent flow across length scales. For this reason, we cannot ignore inertial effects and calculate drag forces as:

$$F_{drag} = \frac{1}{2} \rho v_{max}^2 C_d A \quad (R1.4)$$

Where ρ is the fluid density, v is the velocity, C_d is the drag coefficient (between 0.02 – 0.10 calculated from COMSOL simulations), and A is the area.

Calculations

If we equate $F_{prop} - F_{drag} = 0$, we can calculate the **maximum velocity** v_{max} as:

$$v_{max} = \sqrt[3]{\frac{8 J_{SRT} k_{HFIP}}{\rho C_d l}} \quad (R1.5)$$

The model is validated using experimental results (**Table R1.2**), showing satisfactory agreement with experimentally measured speed of protein motors across different length scales.

Table R1.2. Motor speed across length scales (experimental vs. model)

l (mm)	v_{max} (mm/s) (experimental)	v_{max} (mm/s) (model)
0.1	408 ± 62	465
0.3	385 ± 43	375
0.5	341 ± 37	336
1	250 ± 30	277
3	216 ± 14	225
5	157 ± 28	195
10	148 ± 52	168

Laminar vs turbulent regime discussion

Only two motors ($l = 100 \mu\text{m}$ and $300 \mu\text{m}$) exhibited laminar flow in COMSOL simulations out of seven analyzed motors. If we consider viscous drag for small length scale motors, we can calculate the drag force from Stoke's drag equation (valid at very low Reynolds number):

$$F_{viscous\ drag} = b \eta v l \quad (R1.6)$$

Where η is the fluid dynamic viscosity, v is motor velocity, l is characteristic length scale of the motor, and the b constant comes from a shape factor calculated in COMSOL for laminar flow (for these motors, $b \approx 2$). If we equate $F_{prop} - F_{viscous\ drag} = 0$, we obtain a maximum velocity of:

$$v_{max\ (laminar)} = \sqrt{\frac{2 J_{SRT} k_{HFIP}}{b \eta}} \quad (R1.7)$$

Figure R1.2: motor maximum speed v_{max} as function of characteristic length scale l , considering inertial or viscous drag in calculations.

The $v_{max(laminar)}$ term is independent of length scale, but it is only valid for the two motors exhibiting laminar flow ($l = 100 \mu\text{m}$ and $300 \mu\text{m}$). For these two motors, we calculate a speed of $v_{max(laminar)} = 380 \text{ mm/s}$. We plot the calculated velocities considering inertial and viscous drag in **Figure R1.2**. Calculations with inertial drag fit the experimental data well, but seem to slightly overestimate the velocity of $l = 100 \mu\text{m}$ motor (due to underestimation of drag force). Nonetheless, the predicted velocity for $l = 100 \mu\text{m}$ motor still falls within experimental error (standard deviation), and therefore we accept the proposed model (with inertial drag) as valid for all the explored length scales. However, if we were to explore smaller length scales, we would have to ignore inertial effects and consider viscous drag at small Reynolds numbers.

Mobility lifetime

We also analyze the mobility lifetime of protein motors across length scales. As described in the main text, we observe two mobility regimes: an initial continuous mobility regime (where the fuel close to the interface is easily released) followed by an intermittent mobility regime (where the fuel close to the interface is exhausted and it is replenished by internal diffusion from the motor core). Analogous two-mobility regimes have been previously reported in other Marangoni self-propelled systems⁶. We measured the mobility lifetime of protein motors in the continuous locomotion regime and their total lifetime for all explored length scales (**Figure R1.3**). The mobility lifetime of the initial continuous locomotion, $\tau_{continuous}$, is related to the release of fuel close to the interface (i.e., edge of the motor body), and therefore it is expected to scale with l :

$$\tau_{continuous} \sim A_{contact} b_{continuous} = h l b_{continuous} \quad (\text{R1.8})$$

Where $A_{effective}$ is the effective propulsion area, h is motor thickness, l is characteristic length scale, and $b_{continuous}$ is an experimentally determined parameter ($b_{continuous} = 880 \text{ min/mm}^2$). Note that since the thickness is constant across length scales, $\tau_{continuous}$ scales linearly with l .

The total mobility lifetime τ_{total} (including both continuous and intermittent locomotion regimes) depends on the total amount of fuel stored in the motor, and therefore is dependent on the volume of the motor V :

$$\tau_{total} \sim V b_{total} = h l^2 b_{total}/2 \quad (\text{R1.9})$$

Where b_{total} is an experimentally determined parameter ($b_{total} = 960 \text{ min/mm}^3$). Note that since the thickness is constant across length scales, τ_{total} scales with l^2 .

Figure R1.3: Total and continuous mobility lifetime of protein motors across length scales.

Propulsion modes

We analyzed the locomotion of protein motors and introduced a parameter to predict the motor propulsion mode. We simulated the release of fuel of protein motors with characteristic length scale $l = 100 \mu\text{m}$ to 10 mm in COMSOL. The fuel release was simulated in static conditions (zero fluid speed) after 50 ms in order to evaluate the initial conditions triggering one or another propulsion mode (**Figure R1.4**). Protein films had an initial fuel concentration of 2 kmol/m^3 , diffusion coefficient of HFIP in water $D_{medium} = 3.2 \cdot 10^{-9} \text{ m}^2/\text{s}$ ^{7,8}, and diffusion coefficient of HFIP in the protein $D_{internal} = 7.25 \cdot 10^{-10} \text{ m}^2/\text{s}$ ⁹⁻¹¹. We can observe that, due to their design, small-scale motors ($l = 100 \mu\text{m}$ to 1 mm) generate an anisotropic concentration profile, with an increased local concentration in the posterior cavity. Such anisotropic concentration profile results in an anisotropic surface tension gradient that generates a forward Marangoni propulsive force. On the other hand, larger motors ($l = 3 \text{ mm}$ to 10 mm) have an isotropic concentration profile, with a homogeneous increase in fuel concentration along the motor contact line. In this case, the preferred direction for propulsion is perpendicular to the long axis, hence lateral propulsion⁴.

Figure R1.4: Fuel release simulations in COMSOL for protein motors from $l = 100 \mu\text{m}$ to 10 mm at 50 ms .

In order to quantitatively analyze the propulsion modes, we analyzed the fuel concentration profile in two directions: along the posterior cavity (for forward propulsion) and perpendicular to the posterior leg (for lateral propulsion). In **Figure R1.5a** we plotted the fuel concentration along both directions over the normalized distance from the motor contact line, r/l . We can observe that the concentration follows an exponential decay with increasing distance away from the motor contact line. While larger motors exhibit very similar, sharp decay profiles for both forward and lateral directions, smaller motors exhibit different decay profiles over longer distance for forward and lateral directions. To quantify this, we calculate the decay constant τ for each exponential decay curve, and introduce the predictive dimensionless parameter $\delta_{propulsion} = \tau_{fwd} / \tau_{lat}$, where τ_{fwd} and τ_{lat} are the decay constants for a given length scale motor in the forward and lateral directions respectively (**Figure R1.5b**). For a $\delta_{propulsion} = 1$ ($\tau_{fwd} = \tau_{lat}$), the fuel profiles along the forward and lateral directions are equal, creating an isotropic concentration gradient. $\delta_{propulsion} < 1$ ($\tau_{fwd} \ll \tau_{lat}$) indicates the release of fuel in the lateral direction is dominant (which is not possible in the current design), and $\delta_{propulsion} > 1$ indicates the release along the forward direction is dominant. $l = 10 \text{ mm}$ motors have $\delta_{propulsion} = 1$, indicating an isotropic release (the leg separation is too big to create an increase in concentration in the posterior cavity). This results in lateral propulsion due to Marangoni forces perpendicular to the long axis of the motor⁴. As l decreases, the leg separation is smaller, increasing the concentration in the posterior cavity and consequently increasing $\delta_{propulsion}$. For $1.0 < \delta_{propulsion} < 1.25$, the anisotropy in the concentration gradient around the motor is not strong enough to guarantee continuous propulsion in the forward direction, and the motor alternates between forward and lateral propulsion. For $\delta_{propulsion} > 1.25$, the release of fuel in the posterior cavity is dominant ($\tau_{fwd} \gg \tau_{lat}$) and causes an anisotropic concentration strong enough to propel the motor forward. The prediction of propulsion modes with $\delta_{propulsion}$ agrees with our experimental observations (**Figure S10**), and therefore we accept it as a valid design parameter for programmable locomotion of our protein self-propelled motors.

Figure R1.5: a) Fuel concentration along the forward and lateral directions for protein motors from $l = 100$ μm to 10 mm at 50 ms as function of the normalized distance from the motor contact line r/l . b) Prediction of propulsion modes as function of l with $\delta_{propulsion}$ parameter.

One may ask the question that why suddenly the use of protein molecules may create such a high surface tension gradient which may lead to such high speed propulsion? Or it has some other origin? In that case, the proposed mechanism may be challenged.

This is a very important point, and we have revised our manuscript to provide a clear answer to why the motors have such good performance. The outstanding performance and mobility lifetime of the protein motors is due to the combination of: a) low surface tension of HFIP fuel, b) entrapment of fuel inside the protein matrix, and c) the miniaturization of the motors. We describe each feature below:

a) Low surface tension of HFIP.

In **Figure R1.6**, we compare the surface tension of diverse chemical fuels (commonly used in self-propelled systems) in water as function of fuel molar fraction. We can observe that HFIP is advantageous over other chemical fuels for two main reasons: surface tension of pure HFIP (14.7 mN/m) is significantly lower than that of other fuels, and the surface tension at very low concentrations of HFIP is significantly lower than that of other fuels at equivalent concentrations. This low surface tension at very low concentrations has an important implication in the motor locomotion: small amounts of released HFIP fuel (very low local concentrations) will create large surface tension gradients in water, generating stronger Marangoni forces than other fuels at equivalent concentrations. In other words, smaller amounts of HFIP fuel are necessary to generate larger Marangoni forces than other fuels. This translates into very favorable motor metrics (performance output and efficiency), reported in **Figure 4b** and **Table S4**.

Figure R1.6: Surface tension of chemical fuels in water media as function of fuel molar fraction. Methanol¹², ethanol¹², isopropanol¹², acetone¹³, tetrahydrofuran (THF)¹⁴, dimethylformamide (DMF)¹⁵, dimethylsulfoxide (DMSO)¹⁶, and trifluoroethanol (TFE)¹⁷ data taken from their respective bibliography references.

b) Entrapment of fuel

Another important property of HFIP over other chemical fuels is its capability of dissolving SRT proteins (and other aggregated proteins). Most organic solvents do not dissolve or swell the protein, and therefore the amount of fuel absorbed by the protein is very low (resulting in very short mobility lifetimes in the order of a few seconds). However, HFIP can dissolve SRT proteins to high concentrations, allowing for the facile fabrication of protein motors by film casting and laser micromachining (**Figure 1a**). HFIP fuel is entrapped in the protein matrix (approximately 20%, **Figure S2**) and slowly released to the swimming media through a β -sheet nanocrystal network. The entrapment of fuel and its slow release result in very long mobility lifetimes compared to other chemical fuels (**Table R1.3**). Disk protein motors (20 μm in thickness and 5 mm in diameter) with HFIP fuel exhibited 36 ± 2 minutes of steady uninterrupted locomotion. After this continuous locomotion regime, the motors exhibit intermittent locomotion for more than 2 hours. This irregular locomotion is caused by the exhaustion of fuel at the motor edges and the internal diffusion and replenishment of fuel from the motor core to the edges⁶. Regardless of whether one is interested in the steady mobility regime (as it is in our case) or in the total mobility regime, the performance of the SRT/HFIP motors is superior compared to other chemical fuels.

Table R1.3: Chemical fuels in SRT protein disk motors of $\varnothing = 5\text{mm}$. Density, viscosity, surface tension, capability of dissolving SRT proteins, and motor mobility lifetime.

Chemical fuel in SRT protein motors	Density (g/cm ³)	Viscosity (cP)	Surface tension (mN/m)	Dissolves SRT	Mobility lifetime (s)
Methanol	0.792	0.69	22.1	X	28 ± 13
Ethanol	0.789	1.36	33.0	X	27 ± 21
Isopropanol (IPA)	0.786	0.60	22.3	X	16 ± 2
Acetone	0.784	0.39	23.7	X	51 ± 22
Tetrahydrofuran (THF)	0.889	0.52	26.7	X	10 ± 7
Dimethylformamide (DMF)	0.944	0.79	37.1	X	199 ± 73
Chloroform	1.490	0.59	27.2	X	3 ± 2
Dichloromethane (DCM)	1.330	0.43	26.5	X	3 ± 2
Hexane	0.655	0.30	18.4	X	6 ± 5
Hexadecane	0.770	2.38	27.5	X	7 ± 6
Dimethylsulfoxide (DMSO)	1.100	2.00	44.0	✓	171 ± 27
Trifluoroethanol (TFE)	1.325	1.19	22.2	X	309 ± 169
Hexafluoroisopropanol (HFIP)	1.596	1.65	14.7	✓	2140 ± 92 (steady) (> 2 hour total)

c) Miniaturization of the motors.

The design of Marangoni self-propelled motors is usually restricted in size for two reasons: fabrication of the motor and fuel storage. Fabrication of small-scale motors is challenging due to possible damage and defects in the geometry, leading to random and uncontrolled locomotion. In addition, the smaller the motor is, the less amount of fuel it can carry (scales with cubic length scale) and therefore the lower performance and mobility. For this reason, most self-propelled motors are usually limited to length scales from a few millimeters to centimeters. In the case of the protein motors reported here, we can scale down the characteristic length scale of the motors up to 100 μm (smallest feature size 17 μm) without losing directionality control. These small-scale motors are able to perform long-range locomotion even with a small volume of fuel available thanks to the high fuel efficiency. In addition, the small size results in the decrease of drag (scales with l and l^2 for laminar and turbulent flow respectively) opposing the Marangoni propulsive force and an increase of speed. For these reasons, the miniaturization of protein motors pushes the performance (speed, output, and fuel efficiency) beyond previous limitations in Marangoni self-propelled motors.

(iii) Grossly the paper is written in such a manner where the previous works are not very well respected. In fact, many of the seminal contributions of the previous works have been ignored. Further, the paper can also be criticized for heavy self-citation.

We have revised the manuscript and the references accordingly.

(iv) What type of environmental remediation is expected with these motors? Any example case has not been shown here. The size of the motor excludes its potential in the usage of drug deliver and all. The motor may migrate only at the interface and that also for a small amount of time until it releases the protein to create the surface tension gradient. Perhaps a magnetic handle would have been ideal for such a motor for recovery and recharge.

We thank the reviewer for the comments. In order to completely address this issue, we will break down comment (iv) and address each of the individual points below:

What type of environmental remediation is expected with these motors? Any example case has not been shown here.

In this work, we explore the use of protein motors in environmental remediation of copper pollutants. Industrial activities such as mining, processing of ore, fabrication of electronic devices, and use of pesticides have led to the release of heavy metal contaminants into industrial wastewaters, including copper ions. As a result, high concentration of pollutants has not only been found in hydrological resources, but also in soil, fish, and crops, in levels that are not safe for human activity or consumption¹⁸. Therefore, heavy metal water pollution (including copper ion contamination) is an increasing global concern and poses an immediate threat to human and animal life (both terrestrial and aquatic).

Among the new technologies that are being developed to fight this environmental problem, micro- and nanomotors are rising as promising platforms for efficient removal of organic, inorganic, and pathogenic microorganism contaminants¹⁹⁻²¹. Micromotors for heavy metal removal usually are composed of adsorptive materials (typically carbon-based nanomaterials)²² or are functionalized with chelating agents (molecules that bind to metal ions)²³. In the case of our protein-based micromotors, we took advantage of the natural amino acid histidine, present in SRT proteins (~ 10% mol). Histidine are known natural chelators that bind to copper ions (among others), and are commonly used in protein purification processes (histidine-tag sequence modification)²⁴. We evaluated our SRT-based motors for removal of copper ions from a 1 ppm Cu²⁺ aqueous solution, and measured the removal of $26.4 \pm 4.9\%$ using a single protein motor ($l = 10$ mm). Although the remediation efficiency could be further improved (e.g., by decreasing the motor size and using multiple motors simultaneously), these results demonstrate the metal ion sequestration capabilities of our protein motors for potential future environmental remediation applications, and add multifunctionality to our versatile, non-toxic, and self-propelled platform.

The size of the motor excludes its potential in the usage of drug deliver and all.

Mobile microrobotics is an active and growing research area moving towards applications in the medical field²⁵⁻²⁷. In recent years, a wide variety of robots ranging from several millimeters to few micrometers have been developed for medical applications including drug delivery²⁸, minimally invasive surgery²⁹, and diagnostics among others³⁰. The use of medical milli/microrobots has been already demonstrated *in vivo*, for example in active drug delivery in the stomach (including insulin³¹, antibiotics³², and antigens³³) for therapeutics and oral vaccination, or in wireless ophthalmologic operations in the vitreous humour^{34,35}.

In our previous submission, we demonstrated the use of our protein motors as self-propelled vehicles for cargo delivery. In such proof-of-concept demonstration, motors with characteristic length scale $l = 10$ mm navigated the air-liquid interface of a liquid pool (H₂O), and upon the application of a chemical stimulus (acidic environment) they terminated the locomotion, swelled, and released a model drug to their swimming environment.

Admittedly, the size of the motor ($l = 10$ mm) was quite large for possible navigation in the body, and the model drug (crystal violet dye) was used solely for visualization purposes (although crystal violet has been historically used as antibacterial and antifungal, its medical use has been antiquated by modern drugs)³⁶. For this reason, we demonstrate the release of doxorubicin (DOX, a commonly used chemotherapy agent) on stimuli-responsive protein motors with characteristic length scales $l = 500$ μm and $l = 100$ μm (**Figure R1.7**). We doped the motor solution (SRT protein + HFIP) with DOX (50 $\mu\text{g}_{\text{DOX}} / \text{mg}_{\text{protein}}$, 5%) and proceeded with the fabrication method previously described (film casting and laser micromachining). **Figure R1.7a** shows bright-field microscopy images of a 5×5 $l = 500$ μm DOX-loaded motor array. A 10 μL drop of 5% diluted acetic acid was placed on top, partially wetting the array. The motors in contact with the acidic stimulus degrade over time (disruption of the β -sheet protein cross-linking structures), releasing the encapsulated DOX. **Figure R1.7b** shows fluorescence microscopy images (from natural fluorescence of DOX) of 5×5 $l = 500$ μm DOX-loaded motor arrays with H_2O and acidic stimulus droplets. Motors in contact with H_2O are stable and retain the encapsulated DOX, while motors with acidic stimulus degrade over time and release the encapsulated DOX (increasing fluorescence from the droplet). **Figure R1.7c** shows the analogous experiment (degradation H_2O and acidic stimulus) with $l = 100$ μm DOX-loaded motor arrays, with similar results: $l = 100$ μm motors are stable in H_2O , but release the DOX with acidic stimulus.

These additional experiments demonstrate the capability of the protein motors to encapsulate and selectively release not only common dyes but also chemotherapy agents such as DOX (extensively used in cancer therapy). Furthermore, we demonstrate that this is possible in motors across the explored length scales from 10 mm down to 100 μm (i.e., size-independent stimuli-responsive release mechanism). We have included this discussion and figure (**Figure R1.7**) to the Supplementary Information section, and partially incorporated some of the DOX results in the last main figure for a more comprehensive demonstration of the cargo delivery via protein degradation mechanism.

Figure R1.7: Motor degradation and DOX release. a) Array of $l = 500 \mu\text{m}$ DOX-loaded motors with an acidic stimulus (droplet). b) Fluorescence of $l = 500 \mu\text{m}$ and c) $l = 100 \mu\text{m}$ DOX-loaded motors with H₂O and acidic stimulus.

The motor may migrate only at the interface and that also for a small amount of time until it releases the protein to create the surface tension gradient.

Due to the fact that the motor propulsion is based on Marangoni forces, propulsion is inherently restricted to air-liquid or immiscible liquid-liquid interfaces. This limits the potential use of such motors in physiological environments with naturally occurring interfaces like in stomach or lung alveoli^{32,37,38}. This is a current limitation of the protein motors described in this work, and it is acknowledged in the manuscript. However, our protein-based motors can also potentially work fully immersed in bulk fluid by integrating air cavities and bubbles (engineered interfaces). This would bring design, fabrication, and control challenges that are out of the scope of this manuscript, but will be part of future work in our research group.

Perhaps a magnetic handle would have been ideal for such a motor for recovery and recharge.

Recovery and recharge of self-propelled motors is a useful strategy to extend the lifetime of a motor that has exhausted its fuel, and it has been explored in previous reports⁶. However, it does not represent a practical advantage since it requires retrieving the motor from the swimming medium, drying, refueling, and transferring again to the swimming medium. Multiple refueling cycles degrade the motors due to mechanical damage during contact manipulation and transfer⁶. Furthermore, in an *in vivo* setting, retrieving the motor would require additional operation such as localization of the motors, insertion of needles and tools, and motor subtraction³⁴. SRT protein-based motors can be recharged (infused with HFIP fuel) and completely recycled (protein can be dissolved and used in the fabrication of new motors). However, due to the mentioned disadvantages in recovery and recharge approaches, we have alternatively developed a non-contact manipulation method for locomotion after the fuel is exhausted by photothermally-induced Marangoni propulsion. This method offers a non-contact, continuous transition between chemically- and thermally-driven propulsion modes (giving the motors a second operational lifetime). We have added a full paragraph in the manuscript discussing this approach, and included the results in **Figure 6** and **Figure S22**.

Thus, although I liked the paper much, could not recommend it for Nature Communications. Thank you very much for putting up such a beautiful piece of work.

We thank the reviewer again for the kind words and helpful comments. We have included these discussions in the supplementary information and revised the main text accordingly, which will improve the quality of the article. We hope that our revision satisfies the reviewers and meets the standards for publication in *Nature Communications*.

Reviewer #2 (Remarks to the Author):

This paper reported a versatile protein motor made of squid ring teeth protein. The authors discussed in good details the motor propulsion mechanism, the performance, methods to control the properties of the motor, and several interesting alternative implementations and application-relevant studies in environmental cleaning and self-destructing cargo delivery.

The work is generally very well done, the performance of the motor is very impressive. This is a very solid progress in surface tension driven mini-micro robotic agents. The paper also demonstrates the potential applicability, which may interest researchers in related fields.

We thank the reviewer for carefully reading the paper and for providing very helpful comments. We will address the comments on a point-by-point basis below:

One thing that is not clearly discussed in the paper is the lateral motion of the large version of the motor, and the chaotic motion of the smaller version. In theory, the symmetric shape should lead to no motion in lateral direction. So the motion must come from the nonuniformity of the shape/protein concentration etc. The authors should discuss this at least qualitatively.

In theory, a motor with a symmetric shape will release the fuel equally in all directions, generating an isotropic surface tension gradient that produces no Marangoni propulsion. However, in reality, small instabilities (vibrations, flow fluctuations, or simply the action of carefully transferring the motor to the air/liquid interface) can shift the system out of equilibrium and disrupt the theoretical isotropic surface tension gradient. When randomly disrupted by such instabilities, the surface tension gradient becomes anisotropic and generates Marangoni flows that propel the motor/particle in a random direction. This behavior has been previously described by David Gracias' group (Bassik, et al. *Langmuir* 2008 24 21)⁴ and Bartosz Grzybowski's group (Park, et al. *ACS Nano* 2017 11 11)⁶, where they studied the effect of geometry and symmetry in self-propelled Marangoni motors. Specifically, they observed that "rotationally symmetric shapes tended to precess or rotate"⁴, and that symmetric "circular disks performed random motions"⁶. Furthermore, they observed that symmetric motors with rectangular shapes with aspect ratio 2:1 translated perpendicularly to their longer side. The fuel released from the larger surface area on the long sides created a preferential direction for Marangoni propulsion despite the symmetry of the shape⁴.

These observations also apply to our protein/HFIP motor system, and we have taken advantage of it by designing the motor geometry with different locomotion modes. The motors exhibit three locomotion modes depending on the length scale: orbiting motion (lateral propulsion), straight linear motion (forward propulsion), and a combination of the two. We have simulated the fuel release profile of motors with characteristic length of $l = 10$ mm (lateral propulsion) and $l = 1$ mm (forward propulsion) in COMSOL (**Figure R2.1**). $l = 10$ mm motors had a homogeneous increase in fuel concentration along the contact line, resulting in a symmetric theoretical surface tension gradient (the motor "legs" are too separated to create a local fuel concentration increase in the posterior cavity). However, as previously discussed, instabilities can disrupt the surface tension gradient along the contact line and cause Marangoni propulsion in the direction perpendicular to the long side ("legs" of the motor). Because of the design of the motor, the long sides are at a $\sim 21^\circ$ angle from the symmetry axis, causing the orbiting locomotion of the motor. $l = 1$ mm motors had an anisotropic concentration profile, with an increased local concentration in the posterior cavity due to its design. The anisotropic concentration results in an anisotropic surface tension gradient that generates a forward Marangoni propulsive force. Intermediate motors ($l = 3$ mm and $l = 5$ mm) alternate between lateral and forward propulsion in complex trajectories. At this length scales, the motor "legs" are close enough to

generate a slight local concentration increase in the posterior cavity. However, this anisotropy is weak enough that is easily disrupted by surface flow and instabilities.

Of course, it is possible that the discussed instabilities could come from fabrication defects (i.e., geometry defects) and heterogeneity in the motors (i.e., defects in protein/fuel concentration), as suggested by the reviewer. However, due to the robustness of the fabrication method and the reproducibility of the different propulsion modes in motors across length scales, we believe that it is not the case. In the following, we introduce a parameter based on fuel release COMSOL simulations to predict propulsion modes as function of characteristic length scale of the motors. We will discuss the chemical homogeneity of the protein motors and the reproducibility of the fabrication process in the next two following points.

Propulsion modes

We analyzed the locomotion of protein motors and introduced a parameter to predict the motor propulsion mode. We simulated the release of fuel of protein motors with characteristic length scale $l = 100 \mu\text{m}$ to 10 mm in COMSOL. The fuel release was simulated in static conditions (zero fluid speed) after 50 ms in order to evaluate the initial conditions triggering one or another propulsion mode (**Figure R2.1**). Protein films had an initial fuel concentration of 2 kmol/m^3 , diffusion coefficient of HFIP in water $D_{\text{medium}} = 3.2 \cdot 10^{-9} \text{ m}^2/\text{s}$ ^{7,8}, and diffusion coefficient of HFIP in the protein $D_{\text{internal}} = 7.25 \cdot 10^{-10} \text{ m}^2/\text{s}$ ⁹⁻¹¹. We can observe that, due to their design, small-scale motors ($l = 100 \mu\text{m}$ to 1 mm) generate an anisotropic concentration profile, with an increased local concentration in the posterior cavity. Such anisotropic concentration profile results in an anisotropic surface tension gradient that generates a forward Marangoni propulsive force. On the other hand, larger motors ($l = 3 \text{ mm}$ to 10 mm) have an isotropic concentration profile, with an homogeneous increase in fuel concentration along the motor contact line. In this case, the preferred direction for propulsion is perpendicular to the long axis, hence lateral propulsion⁴.

Figure R2.1: Fuel release simulations in COMSOL for protein motors from $l = 100 \mu\text{m}$ to 10 mm at 50 ms.

In order to quantitatively analyze the propulsion modes, we analyzed the fuel concentration profile in two directions: along the posterior cavity (for forward propulsion) and perpendicular to the posterior leg (for lateral propulsion). In **Figure R2.2a** we plotted the fuel concentration along both directions over the normalized distance from the motor contact line, r/l . We can observe that the concentration follows an exponential decay with increasing distance away from the motor contact line. While larger motors exhibit very similar, sharp decay profiles for both forward and lateral directions, smaller motors exhibit different decay profiles over longer distance for forward and lateral directions. To quantify this, we calculate the decay constant τ for each exponential decay curve, and introduce the predictive parameter $\delta_{propulsion} = \tau_{fwd} / \tau_{lat}$, where τ_{fwd} and τ_{lat} are the decay constants for a given length scale motor in the forward and lateral directions respectively (**Figure R2.2b**). For a $\delta_{propulsion} = 1$ ($\tau_{fwd} = \tau_{lat}$), the fuel profiles along the forward and lateral directions are equal, creating an isotropic concentration gradient. $\delta_{propulsion} < 1$ ($\tau_{fwd} \ll \tau_{lat}$) indicates the release of fuel in the lateral direction is dominant (which is not possible in the current design), and $\delta_{propulsion} > 1$ indicates the release along the forward direction is dominant. $l = 10$ mm motors have $\delta_{propulsion} = 1$, indicating an isotropic release (the leg separation is too big to create an increase in concentration in the posterior cavity). This results in lateral propulsion due to Marangoni forces perpendicular to the long axis of the motor⁴. As l decreases, the leg separation is smaller, increasing the concentration in the posterior cavity and consequently increasing $\delta_{propulsion}$. For $1.0 < \delta_{propulsion} < 1.25$, the anisotropy in the concentration gradient around the motor is not strong enough to guarantee continuous propulsion in the forward direction, and the motor alternates between forward and lateral propulsion. For $\delta_{propulsion} > 1.25$, the release of fuel in the posterior cavity is dominant ($\tau_{fwd} \gg \tau_{lat}$) and causes an anisotropic concentration strong enough to propel the motor forward. The prediction of propulsion modes with $\delta_{propulsion}$ agrees with our experimental observations (**Figure S10**), and therefore we accept it as a valid design parameter for programmable locomotion of our protein self-propelled motors.

Figure R2.2: **a)** Fuel concentration along the forward and lateral directions for protein motors from $l = 100$ μm to 10 mm at 50 ms as function of the normalized distance from the motor contact line r/l . **b)** Prediction of propulsion modes as function of l with $\delta_{propulsion}$ parameter.

If possible, the uniformity of the protein concentration of the materials across the sheet (e.g. of Fig. 1c) could be quantitatively analyzed.

We have quantitatively analyzed the composition and homogeneity of the laser-cut protein motors, and we have included the following discussion to supplementary information.

First, the physical homogeneity of the cast protein films was examined. Defects coming from irregularities in the film (e.g., protein impurities and aggregates, microbubbles, thickness variation in areas close to the film edges) were examined by microscopy, and areas with defects were discarded from the fabrication process. Next, careful optimization of the laser micromachining process was required in order to fabricate reproducible protein motors without defects (explained in detail in the next point). After laser micromachining, each motor was individually inspected by microscopy and those with fabrication defects (approximately 5% of the machined motors) were discarded. Therefore, only successfully fabricated protein motors without visible defects (approximately 95% yield) were used for experiments.

Non-visible defects such as chemical composition and homogeneity of individual motors were investigated by infrared spectroscopy across the surface of motors and across motor length scales. **Figure R2.3a** shows FTIR spectra of a $l = 10$ mm motor taken from five different locations. The five spectra are identical to each other, agreeing with previous characterization of SRT protein films. In order to quantitatively evaluate the chemical homogeneity of the motor, the ratio between the protein amide I band (1650 cm^{-1}) and the HFIP fuel band (1178 cm^{-1}) was measured (**Figure R2.3b**). The amide I / HFIP ratio was constant in the different locations, indicating that the fuel composition is constant throughout the motor geometry.

Figure R2.3: analysis of chemical composition and homogeneity throughout a $l = 10$ mm protein motor geometry. a) FTIR spectra of $l = 10$ mm motor from five different locations. b) Amide I / HFIP band ratio is constant throughout the motor geometry.

In addition, we analyzed the chemical composition and homogeneity of protein motors across length scales. **Figure R2.4a** shows the identical FTIR spectra of protein motors with characteristic length scale $l = 10$ mm to $l = 100\ \mu\text{m}$. The chemical composition and homogeneity was evaluated by measuring the amide I / HFIP band ratio (**Figure R2.4b**), indicating a constant fuel composition in motors across length scales.

Figure R2.4: analysis of chemical composition and homogeneity of protein motors across length scales. a) FTIR spectra of characteristic length scale $l = 10$ mm to $l = 100$ μm motors. b) Amide I / HFIP band ratio is constant across motor length scales.

For the matter of replicability, the authors could give more details in the fabrication steps of the motor. For example, one interesting step is the peeling off, which may deform the structure at a great degree and lead to asymmetry?

Reproducible fabrication of the protein motors is extremely important since any small defect might lead to asymmetry and to uncontrolled propulsion. We have developed a facile, yet robust motor fabrication method based on protein solvent casting and laser micromachining. The fabrication process is summarized in **Figure 1a** (main text and figures), and we are including additional fabrication details in the revised Supplementary Information (**Figure S2**).

SRT proteins were extracted from the suction cups of *Loligo vulgaris* squid and purified as previously reported³⁹. A “motor solution” was prepared by dissolving SRT proteins in HFIP to a concentration of 50 mg/mL (crystal violet dye was added for enhanced visualization). The motor solution was cast (100 μL approximately) on soft polydimethylsiloxane (PDMS) substrates and the HFIP solvent was partially evaporated, leaving a $20 \pm 1\%$ (w/w) of residual HFIP fuel trapped in the protein matrix (**Figure S1, S4**).

The resulting protein films (20 μm in thickness) on PDMS substrates were cut by UV laser micromachining to specified motor designs (low drag profile of G1 ballistic coefficient models, **Figure S3**). Due to the temperature sensitivity of SRT protein thin films, the laser micromachining process required careful optimization to avoid thermal degradation of the protein and to maximize the micromachining resolution (**Figure R2.5**). Protein films were micromachined with a LPKF ProtoLaser U3 scanner-guided UV laser (wavelength 355 nm) with 189 mW power at 50kHz frequency for 50 cycles with 1500 ms delay. Using these optimized parameters, the protein films were cut following the specified CAD model with no visible defects (**Figure R2.5a**). Next, the surrounding protein film was mechanically peeled off with tweezers or a needle tip, leaving the machined motors on the soft PDMS substrates (**Figure R2.5b, Figure R2.5c**). Then, the machined motors were individually peeled off the substrate with tweezers (**Figure R2.5d**), preserving their geometry and mechanical integrity due to their good mechanical properties (~ 0.8 GPa modulus). The motors were then transferred for swimming experiments and characterization. This method was used to

fabricate protein motors with characteristic length scales l ranging from 10 mm down to 100 μm (**Figure 1b**), and it can successfully fabricate large arrays of motors simultaneously (**Figure R2.5e**). Each motor was individually inspected in a microscope, and those with visible fabrication defects (protein impurities, microbubbles, heterogeneities, defects from laser cutting) were discarded (approximately 5% from the total fabrication).

Figure R2.5. Laser micromachining of protein motors. a) Protein motor machined with optimized process parameters. b) Protein motor after peeling off surrounding film. c) Machined motors across length scales (from 10 mm down to 100 μm) on a soft PDMS substrate (same motors as **Figure 1b**). d) Motor ($l = 10$ mm) peeled off the substrate with tweezers without deformation. e) 6x5 array of micromachined $l = 1$ mm motors. f) Defective surface cut due to low cycle repetitions. g) Low-power protein damage due to short cycle delay. h) Deep cut into the substrate due to high cycle repetitions at mid power. i) Protein damage due to high-power machining.

Laser machining with the improper parameters prior to process optimization resulted in defective motors:

- Surface cut (**Figure R2.5f**): machining at low power (189 mW) with low cycle repetition resulted in surface cutting only instead of cutting through the full 20 μm thickness of the films. Since the motors remained connected to the surrounding film through uncut regions, they were peeled off the substrate together with the film. A minimum of 50 cycle repetitions were determined as optimum parameter.
- Protein damage (low-power) (**Figure R2.5g**): machining at low power (189 mW) with small tool delay (i.e., delay between consecutive cycle repetitions) resulted in protein damage. Continuous consecutive cycles did not allow for heat dissipation and thermally degraded the protein (black carbonized region). A cycle delay of 1500 ms was enough to avoid thermal degradation in multiple low-power consecutive cycles.
- Deep cut (**Figure R2.5h**): multiple consecutive cycles (50) at mid-power (1 W) with 1500 ms tool delay cut past the protein film into the PDMS soft substrate, complicating the peeling off of the motors. In addition, it resulted in loss of resolution and geometrical defects.

- Protein damage (high-power) (**Figure R2.5i**): a single cycle at high power (5 W) damaged the protein film via thermal degradation, resulting in severe geometrical defects and loss of mechanical integrity.

We thank the reviewer again for the comments on lateral propulsion modes, chemical homogeneity in our motors, and fabrication process details for reproducibility. We have included these discussions in the supplementary information, which will improve the quality of the revised article. We hope that our revision satisfies the reviewers and meets the standards for publication in *Nature Communications*.

Reviewer #3 (Remarks to the Author):

This article reports on self-propelled protein-based motors made of squid ring teeth with hexafluoroisopropanol (HFIP) as fuel to generate Marangoni propulsion. The material used is of high interest due to several properties: biocompatibility, biodegradability and also due to the structural variation of the material (cross linking related to beta-sheet networking) that can vary according to the preparation method or to the nature of the environment. The versatility of the material is at the origin of the high interest of the present system for which multiple functionalities have been explored (motility response to the chemical properties of the environment, trajectory control by magnetic steering, environment remediation...). The possibilities and efficiency of the system are clearly demonstrated by well designed experiments and deserves publication although there are several important points that should be further analyzed to reach the standards of Nature Communications publications.

We thank the reviewer for carefully reading the paper and for providing very helpful comments. We will address the comments on a point-by-point basis below:

The main question is: why is that system faster and longer lasting than most systems described in the literature? There is finally only one sentence, not bringing much insight, to explain this: We attribute such remarkable metrics to the slow release of the fuel due to entrapment in the protein nanostructure, and to the down scalability (miniaturization) of the motors.

This is a very important point, and we have revised our manuscript to provide a clear answer to why the motors have such good performance. The outstanding performance and mobility lifetime of the protein motors is due to the combination of: a) low surface tension of HFIP fuel, b) entrapment of fuel inside the protein matrix, and c) the miniaturization of the motors. We describe each feature below:

a) Low surface tension of HFIP.

In **Figure R3.1**, we compare the surface tension of diverse chemical fuels (commonly used in self-propelled systems) in water as function of fuel molar fraction. We can observe that HFIP is advantageous over other chemical fuels for two main reasons: surface tension of pure HFIP (14.7 mN/m) is significantly lower than that of other fuels, and the surface tension at very low concentrations of HFIP is significantly lower than that of other fuels at equivalent concentrations. This low surface tension at very low concentrations has an important implication in the motor locomotion: small amounts of released HFIP fuel (very low local concentrations) will create large surface tension gradients in water, generating stronger Marangoni forces than other fuels at equivalent concentrations. In other words, smaller amounts of HFIP fuel are necessary to generate larger Marangoni forces than other fuels. This translates into very favorable motor metrics (performance output and efficiency), reported in **Figure 4b** and **Table S4**.

Figure R3.1: Surface tension of chemical fuels in water media as function of fuel molar fraction. Methanol¹², ethanol¹², isopropanol¹², acetone¹³, tetrahydrofuran (THF)¹⁴, dimethylformamide (DMF)¹⁵, dimethylsulfoxide (DMSO)¹⁶, and trifluoroethanol (TFE)¹⁷ data taken from their respective bibliography references.

b) Entrapment of fuel

Another important property of HFIP over other chemical fuels is its capability of dissolving SRT proteins (and other aggregated proteins). Most organic solvents do not dissolve or swell the protein, and therefore the amount of fuel absorbed by the protein is very low (resulting in very short mobility lifetimes in the order of a few seconds). However, HFIP can dissolve SRT proteins to high concentrations, allowing for the facile fabrication of protein motors by film casting and laser micromachining (**Figure 1a**). HFIP fuel is entrapped in the protein matrix (approximately 20%, **Figure S2**) and slowly released to the swimming media through a β -sheet nanocrystal network. The entrapment of fuel and its slow release result in very long mobility lifetimes compared to other chemical fuels (**Table R3.1**). Disk protein motors (20 μm in thickness and 5 mm in diameter) with HFIP fuel exhibited 36 ± 2 minutes of steady uninterrupted locomotion. After this continuous locomotion regime, the motors exhibit intermittent locomotion for more than 2 hours. This irregular locomotion is caused by the exhaustion of fuel at the motor edges and the internal diffusion and replenishment of fuel from the motor core to the edges⁶. Regardless of whether one is interested in the steady mobility regime (as it is in our case) or in the total mobility regime, the performance of the SRT/HFIP motors is superior compared to other chemical fuels.

Table R3.1: Chemical fuels in SRT protein disk motors of $\varnothing = 5\text{mm}$. Density, viscosity, surface tension, capability of dissolving SRT proteins, and motor mobility lifetime.

Chemical fuel in SRT protein motors	Density (g/cm ³)	Viscosity (cP)	Surface tension (mN/m)	Dissolves SRT	Mobility lifetime
Methanol	0.792	0.69	22.1	X	28 ± 13 s
Ethanol	0.789	1.36	33.0	X	27 ± 21 s
Isopropanol (IPA)	0.786	0.60	22.3	X	16 ± 2 s
Acetone	0.784	0.39	23.7	X	51 ± 22 s
Tetrahydrofuran (THF)	0.889	0.52	26.7	X	10 ± 7 s
Dimethylformamide (DMF)	0.944	0.79	37.1	X	3 ± 1 min
Chloroform	1.490	0.59	27.2	X	3 ± 2 s
Dichloromethane (DCM)	1.330	0.43	26.5	X	3 ± 2 s
Hexane	0.655	0.30	18.4	X	6 ± 5 s
Hexadecane	0.770	2.38	27.5	X	7 ± 6 s
Dimethylsulfoxide (DMSO)	1.100	2.00	44.0	✓	3 ± 1 min
Trifluoroethanol (TFE)	1.325	1.19	22.2	X	5 ± 2 min
Hexafluoroisopropanol (HFIP)	1.596	1.65	14.7	✓	36 ± 2 min (steady) (> 2 hour total)

c) Miniaturization of the motors.

The design of Marangoni self-propelled motors is usually restricted in size for two reasons: fabrication of the motor and fuel storage. Fabrication of small-scale motors is challenging due to possible damage and defects in the geometry, leading to random and uncontrolled locomotion. In addition, the smaller the motor is, the less amount of fuel it can carry (scales with cubic length scale) and therefore the lower performance and mobility. For this reason, most self-propelled motors are usually limited to length scales from a few millimeters to centimeters. In the case of the protein motors reported here, we can scale down the characteristic length scale of the motors up to 100 μm (smallest feature size 17 μm) without losing directionality control. These small-scale motors are able to perform long-range locomotion even with a small volume of fuel available thanks to the high fuel efficiency. In addition, the small size results in the decrease of drag (scales with l and l^2 for laminar and turbulent flow respectively) opposing the Marangoni propulsive force and an increase of speed. For these reasons, the miniaturization of protein motors pushes the performance (speed, output, and fuel efficiency) beyond previous limitations in Marangoni self-propelled motors.

The surface tension properties of the fuel solvent given in the SI should be discussed in regard to solvent commonly used in such systems.

We thank the reviewer for the comments. We have included the surface tension characterization of the HFIP fuel solvent to the Supplementary Information. We have also included the HFIP comparison with other common chemical fuels used in self-propelled motors (discussed in the previous section).

The surface tension of hexafluoroisopropanol (HFIP) fuel was optically measured in a goniometer by the pendant drop method, and it was calculated from analysis of the drop shape using the Young-Laplace equation. We measured the surface tension of HFIP in water in concentrations from 0% to 100% in volume (**Figure R3.2a**), with surface tension ranging from 72.7 to 14.7 mN/m. This allowed us to estimate the local difference in surface tension when HFIP is released from the motor. We measured the difference in surface tension $\Delta\gamma = \gamma_0 - \gamma$ as function of HFIP concentration in the solution (where γ_0 is the surface tension of the water media, and γ is the surface tension of the HFIP solution) (**Figure R3.3b**).

Figure R3.2: Surface tension characterization of hexafluoroisopropanol (HFIP) fuel. a) Surface tension measured in a goniometer by the pendant drop method as function of HFIP concentration (% v/v). b) Difference in surface tension $\Delta\gamma = \gamma_0 - \gamma$ between water media and local HFIP concentration.

For calculation and modeling purposes, a motor releasing HFIP can be considered a moving source, and therefore we can assume that the local concentration of HFIP in the close surroundings of the motor will be very low (contrary to a stationary case). Hence, we can make a linear approximation at low HFIP concentrations to $\Delta\gamma$, providing the calibration parameter k_{HFIP} (used in the model described in **Note S1**).

The parameters used in the model proposed in SI should be more explicit and used to predict more effects than only the size of the motor.

For a better and more comprehensive description of the physics driving our motors, we have revised the model in the Supplementary Information and included a summarized version in the main text.

The proposed model was designed to predict the maximum velocity of the motors based on the type of fuel, material of the motor, swimming medium, design of the motor, and characteristic length scale of the motor. The expression for maximum velocity v_{max} is:

$$v_{max} = \sqrt[3]{\frac{8 J_{SRT} k_{HFIP}}{\rho C_d l}} \quad (R3.1)$$

Where k_{HFIP} is a calibration parameter that depends on the specific type of fuel (in this case, HFIP). k_{HFIP} is experimentally obtained from pendant drop measurements of HFIP/water mixtures in a goniometer (**Figure R3.2**), and gives the surface tension gradient as function of HFIP concentration. J_{SRT} is the diffusion flux of fuel to the swimming medium (moles of released fuel per unit area and per second, measured experimentally) and it depends on the motor material and porosity (in this case, SRT protein). ρ is the swimming media density (in this case, H₂O) and it obviously depends on the physical properties of the fluid. C_d is the drag coefficient of the motor, and it is obtained from COMSOL simulations. Both C_d and the “8” factor in the equation depend on the specific design of the motor (i.e., geometry, **Figure S3**). Last, l is the characteristic length of the protein motors, and defines all other geometric parameters (total length, separation between legs, width of legs, etc.). We use l as our primary input parameter to explore the performance of the motors across length scales, since we can scale up or down all motor dimensional parameters as function of l .

In addition to predicting the maximum velocity of the motors, we have expanded our model to predict the mobility lifetime and the propulsion modes based on our main parameter l , as described below:

Mobility lifetime

We analyzed the mobility lifetime of protein motors across length scales. As described in the main text, we observe two mobility regimes: an initial continuous mobility regime (where the fuel close to the interface is easily released) followed by an intermittent mobility regime (where the fuel close to the interface is exhausted and it is replenished by internal diffusion from the motor core). Analogous two-mobility regimes have been previously reported in other Marangoni self-propelled systems⁶. We measured the mobility lifetime of protein motors in the continuous locomotion regime and their total lifetime for all explored length scales (**Figure R3.3**). The mobility lifetime of the initial continuous locomotion, $\tau_{continuous}$, is related to the release of fuel close to the interface (i.e., edge of the motor body), and therefore it is expected to scale with l :

$$\tau_{continuous} \sim A_{contact} b_{continuous} = h l b_{continuous} \quad (R3.2)$$

Where $A_{effective}$ is the effective propulsion area, h is motor thickness, l is characteristic length scale, and $b_{continuous}$ is an experimentally determined parameter ($b_{continuous} = 880 \text{ min/mm}^2$). Note that since the thickness is constant across length scales, $\tau_{continuous}$ scales linearly with l .

The total mobility lifetime τ_{total} (including both continuous and intermittent locomotion regimes) depends on the total amount of fuel stored in the motor, and therefore is dependent on the volume of the motor V :

$$\tau_{total} \sim V b_{total} = h l^2 b_{total}/2 \quad (R3.3)$$

Where b_{total} is an experimentally determined parameter ($b_{total} = 960 \text{ min/mm}^3$). Note that since the thickness is constant across length scales, τ_{total} scales with l^2 .

Figure R3.3: Total and continuous mobility lifetime of protein motors across length scales.

Propulsion modes

We analyzed the locomotion of protein motors and introduced a parameter to predict the motor propulsion mode. We simulated the release of fuel of protein motors with characteristic length scale $l = 100 \mu\text{m}$ to 10 mm in COMSOL. The fuel release was simulated in static conditions (zero fluid speed) after 50 ms in order to evaluate the initial conditions triggering one or another propulsion mode (**Figure R3.4**). Protein films had an initial fuel concentration of 2 kmol/m^3 , diffusion coefficient of HFIP in water $D_{\text{medium}} = 3.2 \cdot 10^{-9} \text{ m}^2/\text{s}$ ^{7,8}, and diffusion coefficient of HFIP in the protein $D_{\text{internal}} = 7.25 \cdot 10^{-10} \text{ m}^2/\text{s}$ ⁹⁻¹¹. We can observe that, due to their design, small-scale motors ($l = 100 \mu\text{m}$ to 1 mm) generate an anisotropic concentration profile, with an increased local concentration in the posterior cavity. Such anisotropic concentration profile results in an anisotropic surface tension gradient that generates a forward Marangoni propulsive force. On the other hand, larger motors ($l = 3 \text{ mm}$ to 10 mm) have an isotropic concentration profile, with an homogeneous increase in fuel concentration along the motor contact line. In this case, the preferred direction for propulsion is perpendicular to the long axis, hence lateral propulsion⁴.

Figure R3.4: Fuel release simulations in COMSOL for protein motors from $l = 100 \mu\text{m}$ to 10 mm at 50 ms .

In order to quantitatively analyze the propulsion modes, we analyzed the fuel concentration profile in two directions: along the posterior cavity (for forward propulsion) and perpendicular to the posterior leg (for lateral propulsion). In **Figure R3.5a** we plotted the fuel concentration along both directions over the normalized distance from the motor contact line, r/l . We can observe that the concentration follows an exponential decay with increasing distance away from the motor contact line. While larger motors exhibit very similar, sharp decay profiles for both forward and lateral directions, smaller motors exhibit different decay profiles over longer distance for forward and lateral directions. To quantify this, we calculate the decay constant τ for each exponential decay curve, and introduce the predictive parameter $\delta_{propulsion} = \tau_{fwd} / \tau_{lat}$, where τ_{fwd} and τ_{lat} are the decay constants for a given length scale motor in the forward and lateral directions respectively (**Figure R3.5b**). For a $\delta_{propulsion} = 1$ ($\tau_{fwd} = \tau_{lat}$), the fuel profiles along the forward and lateral directions are equal, creating an isotropic concentration gradient. $\delta_{propulsion} < 1$ ($\tau_{fwd} \ll \tau_{lat}$) indicates the release of fuel in the lateral direction is dominant (which is not possible in the current design), and $\delta_{propulsion} > 1$ indicates the release along the forward direction is dominant. $l = 10 \text{ mm}$ motors have $\delta_{propulsion} = 1$, indicating an isotropic release (the leg separation is too big to create an increase in concentration in the posterior cavity). This results in lateral propulsion due to Marangoni forces perpendicular to the long axis of the motor.⁴ As l decreases, the leg separation is smaller, increasing the concentration in the posterior cavity and consequently increasing $\delta_{propulsion}$. For $1.0 < \delta_{propulsion} < 1.25$, the anisotropy in the concentration gradient around the motor is not strong enough to guarantee continuous propulsion in the forward direction, and the motor alternates between forward and lateral propulsion. For $\delta_{propulsion} > 1.25$, the release of fuel in the posterior cavity is dominant ($\tau_{fwd} \gg \tau_{lat}$) and causes an anisotropic concentration strong enough to propel the motor forward. The prediction of propulsion modes with $\delta_{propulsion}$ agrees with our experimental observations (**Figure S10**), and therefore we accept it as a valid design parameter for programmable locomotion of our protein self-propelled motors.

Figure R3.5: a) Fuel concentration along the forward and lateral directions for protein motors from $l = 100$ μm to 10 mm at 50 ms as function of the normalized distance from the motor contact line r/l . b) Prediction of propulsion modes as function of l with $\delta_{propulsion}$ parameter.

The authors attribute the decrease of the maximum speed at low and high pH, in the presence of urea and as a function of temperature (see below) to a higher release of fuel. If higher quantities of fuel are released on the surface, the expected effect would be an acceleration of the motor and not the contrary (the model used by the authors would also predict an increase of the maximum speed if the fuel diffusion flux is increased). Furthermore, in the case of temperature, the effect after the glass transition is rather an increase of the velocity (Figure 3c) than a decrease as claimed by the authors.

The reviewer raises a good point about the performance of the motors in the presence of pH, urea, and temperature stimuli. We have revised text for clarity and included a more complete and detailed explanation.

We differentiate between stimuli that disrupt the protein network (low/high pH and high urea concentration break β -sheet structures) and stimuli that do not (temperature does not break β -sheet structures):

If β -sheet are broken (pH and urea case), the cross-linking structures in the protein network are removed and the protein material can be effectively dissolved. In such case, the protein cannot trap the fuel in its broken network and releases it to the swimming medium (not only the fuel close to the contact line, but the entirety of the fuel, including the motor core). Hence, disrupting the β -sheet cross-linking will terminate the locomotion and reduce the mobility lifetime. Predicting the maximum speed during the protein network disruption is extremely challenging for two main reasons: the system is out of equilibrium, and the parameters are nonlinear and coupled to structural/environmental changes. Not only the flux changes due to stimuli but also the surface tension gradient (e.g., surface tension of acetic acid used to reduce pH is 27.1 mN/m) and the fluid density and viscosity^{40,41}. In addition, the disruption of β -sheet structures can swell and dissolve the protein, causing deformation of the motors and increasing the surrounding medium viscosity (higher drag forces).

In the case of temperature stimulus, β -sheet nanostructures are preserved and the protein network is not disrupted/dissolved. The amorphous protein chains gain flexibility with increasing temperature and allow for a faster diffusion of fuel within and out of the motor. This results in a decrease of mobility lifetime with

temperature due to a faster consumption of the fuel. The fuel flux would increase with temperature, as surface tension, viscosity, and density of water decrease with temperature. This results in an increase of maximum speed with temperature with a plateau above the glass transition temperature. However, the speed and mobility lifetime significantly drop at temperatures higher than 60 °C, probably caused by the evaporation of fuel (boiling point 58.2 °C).

Our proposed model could describe the locomotion of the motors during the disruption of the protein network if all the structural and environmental transitions are considered. However, a quantitative prediction in such situations is out of the scope of this current work, as this phenomenon is only explored as a locomotion termination mechanism (off switch) by self-degradation.

The possibility to use magnetic field to drive the motor is very interesting but the bibliography proposed for a non-original approach is really poor. I would suggest adding more references on that part as for instance A. Ghosh and P. Fischer, Nano Lett. 2009, 9, 2243.

We have added more references on the magnetic propulsion and control of microrobots^{3,42-44}, including the suggested original work by Ghosh & Fischer on magnetic nanopropellers and a more recent publication by the Fischer group where they demonstrate the navigation of magnetic swarms in the vitreous humor of the eye.

The part dedicated to the photothermally induced motion after fuel is exhausted is unclear. What is the dye used ? How are the surface tension gradients generated?

We have modified the manuscript as follows:

An intrinsic limitation of self-propelled motors is the availability of chemical fuel, since motors have no means of doing any work when the fuel is completely exhausted. Previous reports of self-propelled motors explored the possibility of refueling after fuel exhaustion, but it does not represent a practical advantage since it requires retrieving the motor from the swimming medium, drying, refueling, and transferring again to the swimming medium⁶. Furthermore, multiple refueling cycles degrade the motors and deteriorate their performance⁶. In this work, we induced locomotion to the protein motors after the fuel was exhausted via non-contact photothermal propulsion, giving the motors a second lifetime (in addition to their already long mobility).

Light-induced small-scale manipulation via thermocapillary effect has been previously demonstrated by optically heating up the liquid medium, however it usually requires fine positioning, focusing, and regulation of the local heating in order to control thermocapillary convection and particle trajectory⁴⁵⁻⁴⁷. Recently, micro-⁴⁸ and milli-rotors⁴⁹ with a light-absorbing coating have been reported, that heat up and self-generate a local temperature gradient in the surrounding fluid (resulting in Marangoni forces that propel the rotors). Following a similar approach, we photothermally induced locomotion to inactive protein motors with Disperse Red 1 (integrated during the motor fabrication process) by wide-field UV illumination.

Disperse Red 1 is an azobenzene dye that undergoes reversible photoisomerization, converting optical energy into thermal energy by releasing heat into the encapsulating matrix. This approach is commonly used for photothermal actuation in microrobots and actuators by triggering a phase transition in liquid crystal polymer networks and elastomers⁵⁰⁻⁵². Due to the fast photoisomerization of Disperse Red 1 and the high thermal conductivity of SRT proteins¹⁰, the motors were quickly and homogeneously heated upon wide-field UV illumination. When the motor was heated, the temperature of the surrounding swimming fluid increased, creating a temperature gradient along the contact line. Due to the specific geometry of the motor design, there was a local temperature maximum in the fluid at the posterior part of the motor, that

created an anisotropic temperature gradient (**Figure R3.6**). Since the surface temperature of water decreases with temperature⁵³, photothermal heating of the motors caused an anisotropic surface tension gradient (with lowest surface tension in the posterior part of the motor) that generated Marangoni forward propulsion. Infrared thermal images of the locomotion after exhaustion of the fuel showed that only the motor is heated from wide-field illumination while the liquid pool is unaffected, demonstrating that long-range forward locomotion (without chemical fuel) originates from the design of the protein motor. This method offers a non-contact, continuous transition between chemically- and thermally-driven propulsion modes (giving the motors a second operational lifetime).

In addition, we validated this concept computationally (COMSOL simulations) and experimentally (wide-field UV illumination and thermal imaging), and included it to Supplementary Information (**Figure R3.6**).

Figure R3.6: **a)** Heat transfer simulation of a stationary photothermal $l = 1 \text{ mm}$ motor (no flow) shows an anisotropic temperature gradient with maximum temperature at the back of the motor. **b)** Heat transfer simulation of a moving photothermal $l = 1 \text{ mm}$ motor (flow speed 13 mm/s) shows a straight temperature trail behind the motor. **c)** Infrared thermal imaging of a photothermally-propelled $l = 1 \text{ mm}$ motor moving in a straight line at 13 mm/s and leaving a temperature trail behind.

I would also suggest reducing the information provided in each figure. On the printed version they are not readable and some of the information provided could be suppressed or reduced.

We have modified and rearranged the figures so the information is more accessible and easily read, and we have split the figures for better clarity and organization.

We thank the reviewer again for the comments. We have included these discussions in the supplementary information and revised the main text accordingly, which will improve the quality of the article. We hope that our revision satisfies the reviewers and meets the standards for publication in *Nature Communications*.

References

- (1) Maass, C. C.; Krüger, C.; Herminghaus, S.; Bahr, C. Swimming Droplets. *Annu. Rev. Condens. Matter Phys.* **2016**, *7*, 171–193.
- (2) Kwak, B.; Bae, J. Locomotion of Arthropods in Aquatic Environment and Their Applications in Robotics. *Bioinspir. Biomim.* **2018**, *13* (4), 41002.
- (3) Chen, X.; Jang, B.; Ahmed, D.; Hu, C.; De Marco, C.; Hoop, M.; Mushtaq, F.; Nelson, B. J.; Pané, S. Small-Scale Machines Driven by External Power Sources. *Adv. Mater.* **2018**, *30* (15), 1705061.
- (4) Bassik, N.; Abebe, B. T.; Gracias, D. H. Solvent Driven Motion of Lithographically Fabricated Gels. *Langmuir* **2008**, *24* (21), 12158–12163.
- (5) Ikezoe, Y.; Washino, G.; Uemura, T.; Kitagawa, S.; Matsui, H. Autonomous Motors of a Metal–Organic Framework Powered by Reorganization of Self-Assembled Peptides at Interfaces. *Nat. Mater.* **2012**, *11* (12), 1081.
- (6) Park, J. H.; Lach, S.; Polev, K.; Granick, S.; Grzybowski, B. A. Metal–Organic Framework “Swimmers” with Energy-Efficient Autonomous Motility. *ACS Nano* **2017**, *11* (11), 10914–10923.
- (7) Zhang, H.; Duan, W.; Liu, L.; Sen, A. Depolymerization-Powered Autonomous Motors Using Biocompatible Fuel. *J. Am. Chem. Soc.* **2013**, *135* (42), 15734–15737.
- (8) Fioroni, M.; Burger, K.; Mark, A. E.; Roccatano, D. Model of 1, 1, 1, 3, 3, 3-Hexafluoro-Propan-2-ol for Molecular Dynamics Simulations. *J. Phys. Chem. B* **2001**, *105* (44), 10967–10975.
- (9) Pena-Francesch, A.; Jung, H.; Hickner, M. A.; Tyagi, M.; Allen, B. D.; Demirel, M. C. Programmable Proton Conduction in Stretchable and Self-Healing Proteins. *Chem. Mater.* **2018**, *30* (3), 898–905.
- (10) Tomko, J. A.; Pena-Francesch, A.; Jung, H.; Tyagi, M.; Allen, B. D.; Demirel, M. C.; Hopkins, P. E. Tunable Thermal Transport and Reversible Thermal Conductivity Switching in Topologically Networked Bio-Inspired Materials. *Nat. Nanotechnol.* **2018**, *1*.
- (11) Marelli, B.; Brenckle, M. A.; Kaplan, D. L.; Omenetto, F. G. Silk Fibroin as Edible Coating for Perishable Food Preservation. *Sci. Rep.* **2016**, *6*, 25263.
- (12) Vazquez, G.; Alvarez, E.; Navaza, J. M. Surface Tension of Alcohol Water+ Water from 20 to 50. Degree. *C. J. Chem. Eng. Data* **1995**, *40* (3), 611–614.
- (13) Meissner, H. P.; Michaels, A. S. Surface Tensions of Pure Liquids and Liquid Mixtures. *Ind. Eng. Chem.* **1949**, *41* (12), 2782–2787.
- (14) Cheong, W. J.; Carr, P. W. The Surface Tension of Mixtures of Methanol, Acetonitrile, Tetrahydrofuran, Isopropanol, Tertiary Butanol and Dimethyl-Sulfoxide with Water at 25 C. *J. Liq. Chromatogr.* **1987**, *10* (4), 561–581.
- (15) N,N-Dimethylformamide (DMF)-Water Mixture Surface Tension: Datasheet from “Dortmund Data Bank (DDB) – Thermophysical Properties Edition 2014” in SpringerMaterials (https://Materials.Springer.Com/Thermophysical/Docs/Msft_c72c174). Springer-Verlag Berlin Heidelberg & DDBST GmbH, Oldenburg, Germany.
- (16) Markarian, S. A.; Terzyan, A. M. Surface Tension and Refractive Index of Dialkylsulfoxide+ Water Mixtures at Several Temperatures. *J. Chem. Eng. Data* **2007**, *52* (5), 1704–1709.

- (17) Gente, G.; La Mesa, C. Water—Trifluoroethanol Mixtures: Some Physicochemical Properties. *J. Solution Chem.* **2000**, *29* (11), 1159–1172.
- (18) Cai, L.-M.; Xu, Z.-C.; Qi, J.-Y.; Feng, Z.-Z.; Xiang, T.-S. Assessment of Exposure to Heavy Metals and Health Risks among Residents near Tonglushan Mine in Hubei, China. *Chemosphere* **2015**, *127*, 127–135.
- (19) Parmar, J.; Vilela, D.; Villa, K.; Wang, J.; Sánchez, S. Micro-and Nanomotors as Active Environmental Microcleaners and Sensors. *J. Am. Chem. Soc.* **2018**, *140* (30), 9317–9331.
- (20) Jurado-Sánchez, B.; Wang, J. Micromotors for Environmental Applications: A Review. *Environ. Sci. Nano* **2018**.
- (21) Moo, J. G. S.; Pumera, M. Chemical Energy Powered Nano/Micro/Macromotors and the Environment. *Chem. Eur. J.* **2015**, *21* (1), 58–72.
- (22) Vilela, D.; Parmar, J.; Zeng, Y.; Zhao, Y.; Sánchez, S. Graphene-Based Microbots for Toxic Heavy Metal Removal and Recovery from Water. *Nano Lett.* **2016**, *16* (4), 2860–2866.
- (23) Uygun, D. A.; Jurado-Sánchez, B.; Uygun, M.; Wang, J. Self-Propelled Chelation Platforms for Efficient Removal of Toxic Metals. *Environ. Sci. Nano* **2016**, *3* (3), 559–566.
- (24) Arnold, F. H. Metal-Affinity Separations: A New Dimension in Protein Processing. *Nat. Biotechnol.* **1991**, *9* (2), 151.
- (25) Nelson, B. J.; Kaliakatsos, I. K.; Abbott, J. J. Microrobots for Minimally Invasive Medicine. *Annu. Rev. Biomed. Eng.* **2010**, *12*, 55–85.
- (26) Li, J.; de Ávila, B. E.-F.; Gao, W.; Zhang, L.; Wang, J. Micro/Nanorobots for Biomedicine: Delivery, Surgery, Sensing, and Detoxification. *Sci. Robot* **2017**, *2* (4).
- (27) Sitti, M. Miniature Soft Robots—Road to the Clinic. *Nat. Rev. Mater.* **2018**, *3* (6), 74.
- (28) Erkoc, P.; Yasa, I. C.; Ceylan, H.; Yasa, O.; Alapan, Y.; Sitti, M. Mobile Microrobots for Active Therapeutic Delivery. *Adv. Ther.* **2019**, *2* (1), 1800064.
- (29) Khalil, I. S. M.; Tabak, A. F.; Sadek, K.; Mahdy, D.; Hamdi, N.; Sitti, M. Rubbing against Blood Clots Using Helical Robots: Modeling and in Vitro Experimental Validation. *IEEE Robot. Autom. Lett.* **2017**, *2* (2), 927–934.
- (30) Steiger, C.; Abramson, A.; Nadeau, P.; Chandrakasan, A. P.; Langer, R.; Traverso, G. Ingestible Electronics for Diagnostics and Therapy. *Nat. Rev. Mater.* **2018**, *1*.
- (31) Abramson, A.; Caffarel-Salvador, E.; Khang, M.; Dellal, D.; Silverstein, D.; Gao, Y.; Frederiksen, M. R.; Vegge, A.; Hubálek, F.; Water, J. J. An Ingestible Self-Orienting System for Oral Delivery of Macromolecules. *Science* (80-.). **2019**, *363* (6427), 611–615.
- (32) de Ávila, B. E.-F.; Angsantikul, P.; Li, J.; Lopez-Ramirez, M. A.; Ramirez-Herrera, D. E.; Thamphiwatana, S.; Chen, C.; Delezuk, J.; Samakapiruk, R.; Ramez, V. Micromotor-Enabled Active Drug Delivery for in Vivo Treatment of Stomach Infection. *Nat. Commun.* **2017**, *8* (1), 272.
- (33) Wei, X.; Beltrán-Gastélum, M.; Karshalev, E.; Esteban-Fernández de Ávila, B.; Zhou, J.; Ran, D.; Angsantikul, P.; Fang, R. H.; Wang, J.; Zhang, L. Biomimetic Micromotor Enables Active Delivery of Antigens for Oral Vaccination. *Nano Lett.* **2019**.
- (34) Ullrich, F.; Bergeles, C.; Pokki, J.; Ergeneman, O.; Erni, S.; Chatzipirpiridis, G.; Pané, S.;

- Framme, C.; Nelson, B. J. Mobility Experiments with Microrobots for Minimally Invasive Intraocular Surgery. *Invest. Ophthalmol. Vis. Sci.* **2013**, *54* (4), 2853–2863.
- (35) Chatzipirpiridis, G.; Ergeneman, O.; Pokki, J.; Ullrich, F.; Fusco, S.; Ortega, J. A.; Sivaraman, K. M.; Nelson, B. J.; Pané, S. Electroforming of Implantable Tubular Magnetic Microrobots for Wireless Ophthalmologic Applications. *Adv. Healthc. Mater.* **2015**, *4* (2), 209–214.
- (36) Maley, A. M.; Arbiser, J. L. Gentian Violet: A 19th Century Drug Re-emerges in the 21st Century. *Exp. Dermatol.* **2013**, *22* (12), 775–780.
- (37) Gao, W.; Dong, R.; Thamphiwatana, S.; Li, J.; Gao, W.; Zhang, L.; Wang, J. Artificial Micromotors in the Mouse's Stomach: A Step toward in Vivo Use of Synthetic Motors. *ACS Nano* **2015**, *9* (1), 117–123.
- (38) Nkadi, P. O.; Merritt, T. A.; Pillers, D.-A. M. An Overview of Pulmonary Surfactant in the Neonate: Genetics, Metabolism, and the Role of Surfactant in Health and Disease. *Mol. Genet. Metab.* **2009**, *97* (2), 95–101.
- (39) Pena-Francesch, A.; Florez, S.; Jung, H.; Sebastian, A.; Albert, I.; Curtis, W.; Demirel, M. C. Materials Fabrication from Native and Recombinant Thermoplastic Squid Proteins. *Adv. Funct. Mater.* **2014**, *24* (47), 7401–7409.
- (40) Álvarez, E.; Vázquez, G.; Sánchez-Vilas, M.; Sanjurjo, B.; Navaza, J. M. Surface Tension of Organic Acids+ Water Binary Mixtures from 20 C to 50 C. *J. Chem. Eng. Data* **1997**, *42* (5), 957–960.
- (41) Kawahara, K.; Tanford, C. Viscosity and Density of Aqueous Solutions of Urea and Guanidine Hydrochloride. *J. Biol. Chem.* **1966**, *241* (13), 3228–3232.
- (42) Ghosh, A.; Fischer, P. Controlled Propulsion of Artificial Magnetic Nanostructured Propellers. *Nano Lett.* **2009**, *9* (6), 2243–2245.
- (43) Ceylan, H.; Giltinan, J.; Kozielski, K.; Sitti, M. Mobile Microrobots for Bioengineering Applications. *Lab Chip* **2017**, *17* (10), 1705–1724.
- (44) Wu, Z.; Troll, J.; Jeong, H.-H.; Wei, Q.; Stang, M.; Ziemssen, F.; Wang, Z.; Dong, M.; Schnichels, S.; Qiu, T. A Swarm of Slippery Micropropellers Penetrates the Vitreous Body of the Eye. *Sci. Adv.* **2018**, *4* (11), eaat4388.
- (45) Mallea, R. T.; Bolopion, A.; Beugnot, J.-C.; Lambert, P.; Gauthier, M. Laser-Induced Thermocapillary Convective Flows: A New Approach for Noncontact Actuation at Microscale at the Fluid/Gas Interface. *IEEE/ASME Trans. Mechatronics* **2017**, *22* (2), 693–704.
- (46) Mallea, R. T.; Piron, D.; Bolopion, A.; Lambert, P.; Gauthier, M. Thermocapillary Convective Flows Generated by Laser Points or Patterns: Comparison for the Noncontact Micromanipulation of Particles at the Interface. *IEEE Robot. Autom. Lett.* **2018**, *3* (4), 3255–3262.
- (47) Hu, W.; Ishii, K. S.; Ohta, A. T. Micro-Assembly Using Optically Controlled Bubble Microrobots. *Appl. Phys. Lett.* **2011**, *99* (9), 94103.
- (48) Maggi, C.; Saglimbeni, F.; Dipalo, M.; De Angelis, F.; Di Leonardo, R. Micromotors with Asymmetric Shape That Efficiently Convert Light into Work by Thermocapillary Effects. *Nat. Commun.* **2015**, *6*, 7855.
- (49) Wang, W.; Liu, Y.; Liu, Y.; Han, B.; Wang, H.; Han, D.; Wang, J.; Zhang, Y.; Sun, H. Direct Laser Writing of Superhydrophobic PDMS Elastomers for Controllable Manipulation via

- Marangoni Effect. *Adv. Funct. Mater.* **2017**, 27 (44), 1702946.
- (50) Palagi, S.; Mark, A. G.; Reigh, S. Y.; Melde, K.; Qiu, T.; Zeng, H.; Parmeggiani, C.; Martella, D.; Sanchez-Castillo, A.; Kapernaum, N. Structured Light Enables Biomimetic Swimming and Versatile Locomotion of Photoresponsive Soft Microrobots. *Nat. Mater.* **2016**, 15 (6), 647.
- (51) Lahikainen, M.; Zeng, H.; Priimagi, A. Reconfigurable Photoactuator through Synergistic Use of Photochemical and Photothermal Effects. *Nat. Commun.* **2018**, 9 (1), 4148.
- (52) Li, Q. *Photoactive Functional Soft Materials: Preparation, Properties, and Applications*; John Wiley & Sons, 2018.
- (53) Dortmund Data Bank. Surface Tension of Water
http://www.ddbst.de/en/EED/PCP/SFT_C174.php.

REVIEWERS' COMMENTS:

Reviewer #1 (Remarks to the Author):

The authors have addressed all the issues raised in my previous review. I recommend the work to be published in its present form.

Reviewer #2 (Remarks to the Author):

The revised version and the response letter answered the comments well, even though no theoretical justification was given for the lateral motion of the motor. The current version can be published.

Reviewer #3 (Remarks to the Author):

The authors have taken into account all the remarks and their answers are really convincing. The article now reaches my expectations and I think it is an important contribution to self-propelled motors bringing these kind of systems closer to applications. I now recommend the article for publication.